# Revisiting Out-of-distribution Robustness in NLP: Benchmark, Analysis, and LLMs Evaluations

**Lifan Yuan[1], Yangyi Chen[2], Ganqu Cui[1], Hongcheng Gao[3],**
**Fangyuan Zou[4], Xingyi Cheng[4], Heng Ji[2], Zhiyuan Liu[1]\*, Maosong Sun[1]\***
[1] NLP Group, DCST, IAI, BNRIST, Tsinghua University, Beijing
[2] University of Illinois Urbana-Champaign
[3] University of Chinese Academy of Sciences [4] Tencent
`lievanyuan173@gmail.com`

## Abstract

This paper reexamines the research on out-of-distribution (OOD) robustness in the field of NLP. We find that the distribution shift settings in previous studies commonly lack adequate challenges, hindering the accurate evaluation of OOD robustness. To address these issues, we propose a benchmark construction protocol that ensures clear differentiation and challenging distribution shifts. Then we introduce **BOSS**, a **B**enchmark suite for **O**ut-of-distribution robustne**SS** evaluation covering 5 tasks and 20 datasets. Based on BOSS, we conduct a series of experiments on pretrained language models for analysis and evaluation of OOD robustness. First, for vanilla fine-tuning, we examine the relationship between in-distribution (ID) and OOD performance. We identify three typical types that unveil the inner learning mechanism, which could potentially facilitate the forecasting of OOD robustness, correlating with the advancements on ID datasets. Then, we evaluate 5 classic methods on BOSS and find that, despite exhibiting some effectiveness in specific cases, they do not offer significant improvement compared to vanilla fine-tuning. Further, we evaluate 5 LLMs with various adaptation paradigms and find that when sufficient ID data is available, fine-tuning domain-specific models outperform LLMs on ID examples significantly. However, in the case of OOD instances, prioritizing LLMs with in-context learning yields better results. We identify that both fine-tuned small models and LLMs face challenges in effectively addressing downstream tasks. The code is public at `https://github.com/lifan-yuan/OOD_NLP`.

## 1 Introduction

Pretrained language models (PLMs) have excelled in downstream tasks and gained widespread adoption [24, 60]. However, existing evaluation often assumes independent and identically distributed (i.i.d) condition [94, 92], which is often violated in real-world scenarios, highlighting the crucial problem of out-of-distribution (OOD) robustness in NLP models. In this paper, we first revisit the evaluation of PLMs through an examination of evaluation benchmarks. Thereafter, we delve into the ID-OOD performance correlation of fine-tuned models by adopting various model scales, training steps, available training samples, and tunable parameters. Finally, we conduct extensive evaluations of current robustness-enhanced methods and large language models (LLMs).

**Definition.** There exist multiple definitions of OOD in literature [2, 115], and we define distribution shifts considered in this paper from two perspectives. Firstly, [2] classifies distribution shifts into "semantic shift" and "background shift". Our use of "out-of-distribution" aligns with the concept of

---

\*Corresponding Author.

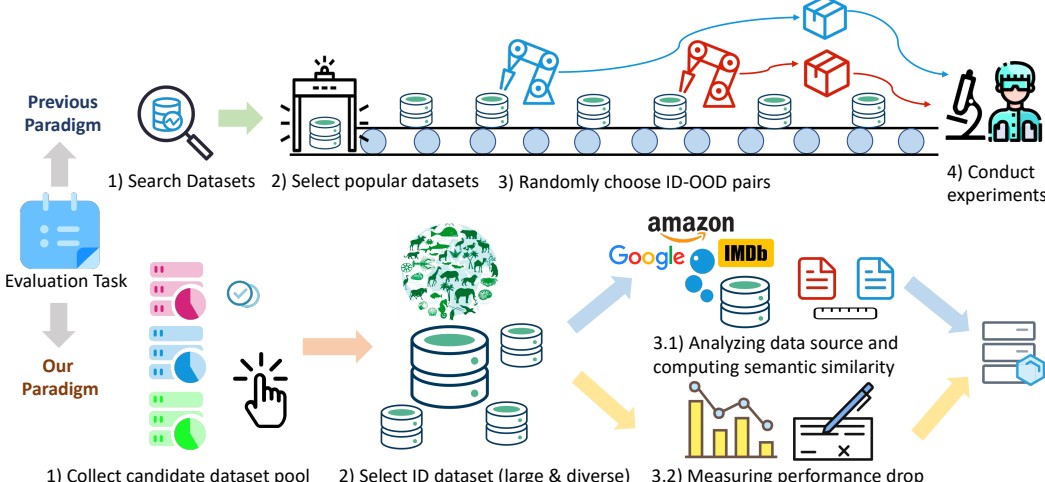

Figure 1: The comparison of previous work and our protocol on dataset selection of OOD benchmarks.

"background shift", which involves changes in the domain or style of the text while preserving the semantic content. Secondly, [115] formally defines three types of distribution shifts: covariate shift, label shift, and concept shift. In our work, we mainly focus on the combination of covariate shift and concept shift. This indicates that the model needs to generalize well to different input features (a.k.a, covariate shift) and adapt to variations in the underlying concepts within the data (a.k.a, concept shift).

**Benchmark.** Our study begins by surveying existing literature on OOD robustness in NLP (Table 8). We observe a lack of standardized OOD benchmark suites tailored for NLP since there is no single uniform set of datasets for evaluation, resulting in the adoption of heuristic and popularity-based dataset selection strategies in previous work [39, 109, 107]. This approach suffers from two main drawbacks: (1) The selected OOD datasets may come from similar distributions as the ID dataset, reducing the OOD evaluation to ID settings and thus hindering rigorous OOD robustness evaluation; (2) The challenge stemming from distribution shifts is limited, deviating from the expectation for difficulty posed by benchmark construction principles [8] and potentially leading to an overestimation of the OOD robustness of language models. As a result, current OOD robustness benchmarks may inadequately assess NLP models.

To address the aforementioned concerns, we establish a protocol as shown in Figure 1, consisting of three fundamental principles, for selecting both ID and OOD datasets: (1) ID datasets should be large and diverse for comprehensive knowledge; (2) Selection for OOD datasets should prioritize distinct distributions and dissimilarity, regarding text sources and semantics; (3) Challenging distribution shifts should be prioritized based on performance degradation, to ensure that the benchmark stands the test of time [8]. Based on the protocol, we compile **BOSS**, a more holistic and challenging NLP **B**enchmark suite for **O**OD robustne**SS** evaluation. Unlike existing benchmarks which only consider single task types such as classification [39, 107] or reading comprehension [109], BOSS covers a wider range of task formats, including sentiment analysis, toxic detection, and natural language inference for classification, name entity recognition for structured prediction, and extractive question answering for reading comprehension. We establish one ID and three corresponding OOD datasets for each task.

**Analysis.** We recognize the lack of analysis of models' learning behaviors regarding ID performance and OOD generalization in the field of NLP, hindering the development and understanding of OOD robustness. Thus, we investigate the correlation between the performance on ID and OOD datasets using the BOSS benchmark. To regulate the ID performance, we manipulate four related factors, i.e. model scale, training steps, available training samples, and tunable parameters. Three typical categories of ID-OOD correlation are observed, namely, monotonic linear positive correlation, monotonic piecewise linear positive correlation, and non-monotonic V-shaped correlation (see Figure 2). We discuss the potential reasons for the causes of these identified correlations in section 3.

**Evaluations.** After examining the learning mechanism of PLMs in vanilla fine-tuning, we scrutinize their performance with existing robustness-enhanced methods and then proceed to prevalent LLMs. Due to the absence of a standard benchmark, previous evaluations of existing methods can be

imprecise and thus misleading the estimations of progress in this field. Moreover, given the increasing focus on LLMs [9, 89] in NLP research, it is essential to evaluate their effectiveness in handling OOD challenges and explore the efficacy of different adaptation paradigms.

For robustness-enhanced methods, we evaluate five representative methods [99] on BOSS. Our main observation is that vanilla fine-tuning (a.k.a, empirical risk minimization) remains a strong baseline, while certain methods may slightly improve OOD performance in some cases. We further evaluate various LLMs and adaptation paradigms. We consider three recent prevailing LLMs, namely LLaMA [89], OpenAI text-davinci-003 [9], and OpenAI gpt-3.5-turbo. We include two relatively smaller models T0-3B [81] and T5-3B [77] for comparison. We apply zero-shot inference, in-context learning, few-shot fine-tuning, and full-data fine-tuning to one or multiple models. Through our experiments, we find that when provided with enough training data, fine-tuning domain-specific models remain the preferable choices for handling ID examples, while leveraging LLMs with in-context learning is superior for tackling OOD instances. In addition, we observe that the impact of in-context learning on generalization ability varies across models. We provide more detailed discussions in section 4.2.

## 2 BOSS Benchmark

### 2.1 Motivation

NLP models should exhibit robustness across diverse distributions to ensure reliable applications. To achieve this, a standardized and recognized evaluation benchmark for OOD robustness is imperative. However, previous efforts in constructing benchmarks have predominantly relied on random selections and dataset popularity, lacking a systematic design [39, 109, 107]. Two deficiencies are thus identified: (1) Dataset similarity, as exemplified by the SST and IMDb datasets for sentiment analysis [83, 64], which share movie reviews and exhibit high semantic similarity (see Table 2). This blurs the line between ID and OOD evaluation, hindering rigorous assessment of OOD robustness; (2) Limited distribution shift challenges, exemplified by the high accuracy of a model trained on Amazon [65] when tested on IMDb (see Table 3). However, the significant performance drop on our considered Dynasent [75] suggests that OOD robustness still remains a critical problem in the sentiment analysis task. Thus, there is a need for universally applicable challenges across all dataset selections [8].

### 2.2 Protocol to Construct OOD benchmark.

We aim to establish a standard benchmark for rigorous evaluation of OOD robustness in NLP. To address the above issues, we first survey and gather existing candidate datasets from `Paperswithcode`[2], `Kaggle`[3], and `ACL Anthology`[4] websites. We consider the release date and public availability of datasets. Then we carefully examine three criteria to determine the ID and corresponding OOD datasets. The first criterion focuses on the ID dataset selection, and the other two criteria are proposed for OOD datasets, targeting the two issues in previous work, respectively.

**The ID dataset should provide sufficient knowledge for models to handle the task.** ID dataset should encompass comprehensive task-level knowledge [44], enabling models to grasp the underlying rationale necessary for task completion. Alternatively, if the model exclusively learns biased features, it may struggle to adapt to other features during distribution shifts. To this end, it is necessary for the ID datasets to possess the following characteristics: (1) Sufficiently large size; (2) Diversity, which is achieved through collection from multiple sources or the inclusion of several subtypes (i.e., styles, topics, levels of formality, et al). Our intuition is in line with [87], which demonstrates that training on large and diverse datasets improves the robustness of vision models.

**Datasets within a given task should originate from diverse distributions for a holistic evaluation.** We guarantee this through qualitative analysis of data source diversity and quantitative measurement of semantic similarity using SimCSE [31]. To avoid overlap, we select at most one dataset per text source. Additionally, we ensure OOD datasets in the benchmark exhibit relatively low semantic similarity, and thus enhancing distinctiveness.

---

[2]https://paperswithcode.com/datasets
[3]https://www.kaggle.com
[4]https://aclanthology.org/

**OOD shifts should be challenging to provide an accurate assessment of progress in OOD robustness [8]**. To quantify the challenge, we train a model on the ID dataset and test it on all candidate datasets. Specifically, we tune a T5-large [77] with manual templates on four tasks, except for NER, on which we adopt DeBERTa-large [38] with conventional fine-tuning due to the lack of a standard prompt-based tuning schema for this task. *For this reason, all experiments in this paper follow this choice of model selection.* For each text source, we first discard candidates similar to the ID dataset in semantics. Then, to construct challenging distribution shifts, we prioritize the dataset provoking the most severe performance drop of the ID model and adopt it as the OOD dataset in our benchmark.

## 2.3 Dataset Selection

We take sentiment analysis as a case to demonstrate how we select ID and OOD datasets for each task according to our protocol. The selection process for other tasks can be found in Appendix D.

**Candidate Datasets.** We first collect all sentiment analysis datasets on `Paperswithcode`, `Kaggle`, and `ACL Anthology` as aforementioned. We filter out datasets released before the 2010s, as they are largely resolved with the advent of pre-trained language models [25]. As a result, seven datasets remain as candidates, i.e., Amazon [65], DSC [48], Dynasent [75], IMDb [64], SemEval [70], SST [83], and Yelp [116]. Considering the inconsistency in the number of categories across the datasets, we align them by converting them into a three-class classification setting. See Appendix C.2 for a detailed explanation of the dataset processing procedure.

**Probing Experiments.** According to our protocol, dataset size and text sources are assessed for ID dataset selection. Subsequently, semantic similarity and ID model performance degradation guide OOD dataset selection. To this end, two probing experiments are conducted: (1) Comparing semantic similarity using SimCSE for candidate dataset pairs, and (2) Evaluating the

Table 1: Statistics of sentiment analysis candidate datasets.

| Dataset | Source | # Classes | # Samples | | Avg. Length | |
| --- | --- | --- | --- | --- | --- | --- |
| | | | Train | Test | Train | Test |
| Amazon | Product | 3 | 30,000 | 38,905 | 71.69 | 54.84 |
| DSC | Product | 2 | 92,244 | 11,531 | 132.29 | 130.14 |
| Dynasent | Adversarial | 3 | 93,553 | 4,320 | 13.19 | 13.83 |
| IMDb | Movie | 2 | 25,000 | 25,000 | 233.79 | 228.53 |
| SemEval | Twitter | 3 | 6,000 | 20,622 | 19.44 | 19.62 |
| SST | Movie | 3 | 4,004 | 1,067 | 18.78 | 18.75 |
| Yelp | Product | 3 | 30,000 | 30,000 | 132.49 | 131.62 |

performance of the selected ID model. In the first experiment, for better semantic representation, we resort to the best SimCSE model provided by [31], a supervised RoBERTa-large [60]. We load the model checkpoint from Huggingface[5]. For each dataset, we first encode each sample into a continuous embedding and then average the embeddings across the dataset to obtain a centroid representation of the dataset. Finally, we calculate the cosine similarity between a pair of centroids as the semantic similarity between two datasets. In the second experiment, we train a T5-large model on the selected ID dataset and evaluate its performance on all the candidate datasets.

**Dataset Selection.** The dataset information and semantic similarities are provided in Table 1 and Table 2, respectively. The text sources of the datasets vary from product reviews, movie reviews, Twitter, and adversarial texts. We observe that datasets originating from the same source tend to exhibit higher SimCSE scores, indicating higher seman-

Table 2: SimCSE scores between each pair of datasets regarding the sentiment analysis task.

| Train \| Test | Amazon | DSC | Dynasent | IMDB | SemEval | SST | Yelp |
| --- | --- | --- | --- | --- | --- | --- | --- |
| Amazon | **100** | 86.02 | 57.30 | 36.67 | 24.74 | 33.70 | 49.22 |
| DSC | 86.02 | **100** | 59.15 | 54.55 | 31.70 | 44.40 | 55.45 |
| Dynasent | 57.30 | 59.15 | **100** | 32.69 | 28.17 | 19.68 | 88.99 |
| IMDb | 36.67 | 54.55 | 32.69 | **100** | 46.95 | 84.62 | 39.88 |
| SemEval | 24.74 | 31.70 | 28.17 | 46.95 | **100** | 40.45 | 24.03 |
| SST | 33.70 | 44.40 | 19.68 | 84.62 | 40.45 | **100** | 19.43 |
| Yelp | 49.22 | 55.45 | 88.99 | 39.88 | 24.03 | 19.43 | **100** |

tic similarity. It is worth noting that for IMDb and SST, the widely used ID-OOD dataset pair in sentiment analysis [39, 107], the SimCSE score demonstrates one of the highest levels among dataset pairs. This reinforces the first deficiency of previous benchmarks, where dataset pairs have similar semantics and unclear distribution shifts. Hence, in contrast to existing practices, our benchmark construction considers only one dataset from each source.

For the ID dataset selection, we first exclude DSC and IMDb since they are binary classification datasets, on which the trained model cannot tackle the unseen class `neutral`. For dataset size,

---

[5]https://huggingface.co/princeton-nlp/sup-simcse-roberta-large

SemEval and SST are disregarded due to their limited number of samples per class (less than 10k). Among the remaining datasets, Amazon is chosen as the ID dataset for sentiment analysis as it encompasses reviews from 29 distinct product categories, offering greater diversity than Yelp.

For OOD datasets selection, we train a T5-large model on the ID dataset (i.e., Amazon) and evaluate it on all candidate datasets, as illustrated in Table 3. We include Dynasent and SemEval in

Table 3: The OOD performance of the T5-large when trained on the Amazon dataset.

| Train | Test | Amazon | DSC | Dynasent | IMDb | SemEval | SST | Yelp |
|---|---|---|---|---|---|---|---|---|
| Amazon | | **90.94** | 95.63 | 47.38 | 92.69 | 49.90 | 75.16 | 89.25 |

the benchmark suite due to the following reasons: (1) They are the sole adversarial and Twitter datasets available, (2) They demonstrate low semantic similarity, and (3) They exhibit a notable performance degradation, making them crucial for evaluation. For movie reviews, SST is prioritized due to lower SimCSE scores compared to IMDb and larger performance drop of the ID model. Eventually, this yields three distinct and challenging distribution shifts in the sentiment analysis task: Amazon → (DynaSent, SemEval, SST).

## 2.4 BOSS

Based on the aforementioned protocol, we introduce BOSS, an NLP **b**enchmark suite for **OO**D robustne**ss** evaluation. BOSS comprises five essential NLP tasks: sentiment analysis (SA), toxic detection (TD), natural language inference (NLI), name entity recognition (NER), and ex-

Table 4: The datasets included in the BOSS benchmark. Corresponding abbreviations are shown in brackets.

| Task | ID Dataset | | OOD Datasets | |
|---|---|---|---|---|
| SA | Amazon (AZ) | Dynasent (DS) | SemEval (SE) | SST (SST) |
| TD | Civil Comments (CC) | AdvCivil (AC) | Implicit Hate (IH) | ToxiGen (TG) |
| NLI | MNLI (MN) | ANLI (AN) | ContractNLI (CN) | WANLI (WN) |
| NER | FewNerd (FN) | CoNLL (CoNLL) | E-NER (ENER) | WNUT (WNUT) |
| EQA | SQuAD (SQuAD) | AdvQA (AQA) | NewsQA (NQA) | SearchQA (QA) |

tractive question answering (EQA). These tasks represent diverse practical applications and provide comprehensive coverage for evaluating models' capabilities, from aspects of classification, structured prediction, and extraction. Each task in the benchmark includes one ID dataset and three associated OOD datasets (see Table 4).

**Sentiment Analysis. Amazon** [65] contains reviews of 29 different categories of products from the Amazon website. **DynaSent** [75] first identifies naturally challenging sentences from several existing datasets, and then creates adversarial sentences with a human-and-model-in-the-loop annotation approach. **SemEval** [70] is a three-class sentiment analysis dataset focusing on tweets. **SST** [83] consists of sentence-level movie reviews from the Rotten Tomatoes website.

**Toxic Detection. Civil Comments** [6] contains public comments on the Civil Comments platform, with users from diverse groups and various subtypes of toxic texts. **AdvCivil**, a new toxic dataset introduced in this paper, is generated from Civil Comments by textual adversarial attacks in an automated model-in-the-loop adversarial pipeline. Please refer to Appendix C.1 for details. **Implicit Hate** [29] contains toxic tweets in both explicit and implicit forms. The latter can circumvent keyword-based toxic detection systems. **ToxiGen** [36] is synthesized by GPT-3 [9], covering several types of subtly and implicitly toxic texts on 13 minority groups.

**Natural Language Inference. MNLI** [102] provides ten different categories of written and verbal sentence pairs, with diverse styles, topics, and levels of formality. **ANLI** [73] is an adversarial dataset collected in a human-and-model-in-the-loop approach, where each premise mainly comes from Wikipedia and the hypothesis is generated by human adversaries. **ContractNLI** [49] considers each contract as a premise and holds a fixed set of hypotheses throughout the dataset. **WANLI** [59] is synthesized by GPT-3 [9], each example containing challenging patterns identified in MNLI.

**Name Entity Recognition. Few-NERD** [26], the arguably largest dataset for NER, labels about 188k Wikipedia sentences into eight coarse-grained entity types. **CoNLL** [88] takes stories from Reuters news, containing four basic entity types. **E-NER** [3] is based on legal text. We use the four-category version in this paper, which treats all legal entities as miscellaneous ones. **WNUT** [23] collects training data from Twitter and mines test data from Reddit, StackExchange, Twitter, and YouTube, containing six coarse-grained entity types in Few-NERD.

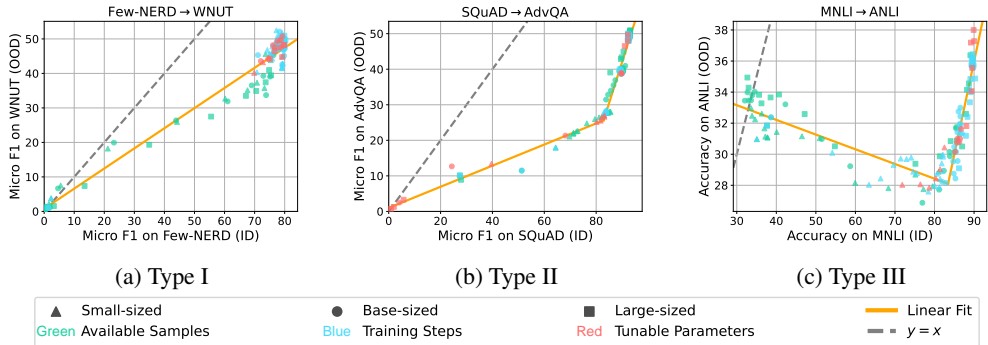

(a) Type I        (b) Type II        (c) Type III

▲ Small-sized    ● Base-sized    ■ Large-sized    — Linear Fit
Green Available Samples    Blue Training Steps    Red Tunable Parameters    -- $y = x$

Figure 2: Three representative correlations between ID-OOD performance: (a) Type I (monotonic linear positive correlation) indicates consistent linear improvement of OOD performance with increasing ID performance. (b) Type II (monotonic piecewise linear positive correlation) exhibits accelerated OOD performance growth after a turning point. (c) Type III (non-monotonic V-shaped correlation) shows an initial negative correlation, followed by a positive correlation after a turning point. The $r^2$ value in Figure (a) is 0.9677, and the values of the left and right fits in Figure (b) are 0.9553 and 0.9396 whereas in Figure (c) are 0.7690 and 0.8124 respectively.

**Extractive Question Answering. SQuAD** [78] constructs question-answer pairs based on Wikipedia passages. **AdversarialQA** [4] composes adversarial questions for contexts in SQuAD in a human-and-model-in-the-loop procedure, similar to ANLI. **NewsQA** [90] writes questions for CNN news articles, each of which requires reasoning to answer, rather than relying solely on word overlap and textual entailment. **SearchQA** [28] adopts a reverse construction pipeline, employing the Google search engine to retrieve relevant contexts for each question-answering pair from the J!Archive website.

## 3 Analysis of OOD Robustness

Despite OOD robustness in NLP has been extensively studied [43], a potential concern pertains to the usage of nonstandard benchmarks, as discussed in Section 2, resulting in inaccurate conclusions. To address this issue, we conduct a series of empirical analyses and evaluations to gain in-depth insights into OOD robustness in NLP. Previous research primarily concentrates on method comparisons without delving into models' learning behaviors. Therefore, we first analyze the models' learning mechanism by assessing the correlation between ID and OOD performance.

**Setting.** We assess the correlation between ID and OOD performance across various conditions. We manipulate the ID performance of models by varying their scale, training steps, available training samples, and tunable parameters. Further implementation details can be found in Appendix E.1.1.

**Results.** We observe that the correlation between ID and OOD performance on datasets of the five tasks is inconsistent, but can be broadly categorized into three types (see Figure 2): monotonic linear positive correlation (**Type I**), monotonic piecewise linear positive correlation (**Type II**), and non-monotonic V-shaped correlation (**Type III**). We also identify an exceptional case in Figure 3, which does not fall into any of the three categories. The full results are shown in Figure 4.

**Type I.** This is the most prevalent type of correlation observed across all ID-OOD pairs for sentiment analysis, name entity recognition, and the majority for toxic detection. As shown in Figure 2a, in this category, OOD performance is positively and linearly correlated with ID performance, indicating that the task knowledge learned on source distribution can be effectively generalized to other distributions. This observation is consistent with results in the computer vision domain [68], which shows that OOD performance is linearly correlated with ID performance across various model architectures, hyperparameters, training dataset size, and training duration. However, the slope of the line fitted by the least square method is less steep than the $y = x$ diagram, and it eventually lies below the diagonal, implying that the performance degradation of models under distribution shift will be escalated with the increase of the ID performance.

**Type II.** This category is observed on ID-OOD pairs for extractive question answering. As presented in Figure 2b, the results can be fitted into a polyline, indicating a piecewise linear correlation. The correlation between OOD performance and ID performance is positive and linear, with a notable differ-

ence in the slope before and after the turning point. Specifically, OOD performance demonstrates slow growth until the turning point, after which a minor increase in ID performance yields a substantial improvement in OOD performance. The observed trend may be attributed to the findings of [91], which indicates that models initially capture spurious correlations in ID datasets before acquiring comprehensive task knowledge. Consequently, models prioritize learning these spurious correlations to address the ID task, resulting in minimal improvements on OOD datasets. However, in the later stages of training, models progressively acquire greater task knowledge, leading to improved OOD performance.

**Type III.** The V-shaped fitted lines shown in Figure 2c mainly occurs on ID-OOD pairs of NLI tasks. This pattern is divided into two stages by a turning point in ID performance. In the first stage, OOD performance experiences worsening performance degradation during the distribution shift. However, in the second stage, the ID-OOD correlation becomes positive. This trend resembles the U-shaped scaling law of LLMs observed by [100], thus suggesting a shared explanation. [100] attributes this phenomenon to the "distractor task" in the dataset, apart from the "true task". Medium-capacity models may perform better than low-capacity models on the distractor task, which may harm performance on the "true task". As the model capability increases, it can ignore the distractor task and focus on improving performance on the true task. Here, we identify the distractor task in NLI datasets as detecting the word overlap or other spurious correlations between the premise and hypothesis.

**Outlier.** There is an exceptional case regarding the distribution shift from Civil Comments to AdvCivil (see Figure 3). The figure depicts two distinct lines, both exhibiting a monotonic linear negative correlation. This may stem from the model's increased reliance on spurious correlations and the adversarial nature of AdvCivil. Prior research suggests that models can learn non-robust features, such as spurious correlations, to enhance ID accuracy [44]. However, adversarial samples preserve the semantics of the original texts while introducing perturbations that eliminate spurious correlations. Hence, when the ID model becomes more dependent on spurious correlations during training, its performance degradation on adversarial samples intensifies.

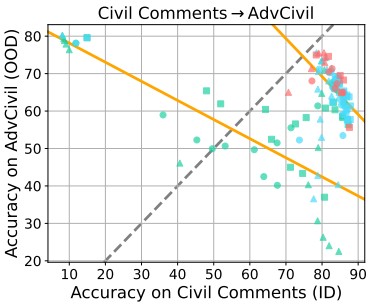

Figure 3: The OOD performance exhibits a negative correlation with ID performance. Refer to Figure 2 for legends.

## 4  Evaluation of OOD Robustness

### 4.1  Robustness-enhanced Methods

After analyzing the learning behavior of PLMs under vanilla fine-tuning, we examine their performance when trained with other methods. Although massive methods have been proposed to improve the robustness of PLMs, their evaluations rely on non-standard benchmarks, which may result in inaccurate evaluations and impede progress clarity. Therefore, in this section, we first conduct extensive experiments to re-evaluate the effectiveness of diverse robustness-enhanced methods.

**Setting.** We consider the categories of robustness-enhanced methods summarized by [99]: data-driven, model and training-based, inductive-prior-based, and causal intervention methods. We select the most representative one from each category for evaluation. Specifically, we choose EDA [101] and FreeLB [118] for data-driven methods, label smoothing [84] and focal loss [58] for model and training-based methods, and model ensemble [16] for inductive-prior-based methods. We do not consider causal intervention methods since they are typically applied to low-resource scenarios. As explained in section 2.3, we apply the above methods to DeBERTa-base models for the NER task and to T5-base models for the other tasks.

**Results.** The results are shown in Table 5, where the mark '-' indicates that a certain method is not applicable to the task. We summarize the findings in the following takeaways:

*Takeaway 1: The vanilla fine-tuning (empirical risk minimization) remains a strong baseline.* Despite existing methods outperforming vanilla fine-tuning on certain datasets like E-NER, WNUT, and NewsQA, they show limited superiority or can potentially harm model performance. Specifically, only FreeLB demonstrates beneficial effects over half of the datasets, standing out as the most

Table 5: The evaluation of robustness-enhanced methods. The results that surpass the vanilla baseline are underlined. We use abbreviations representative of the datasets to conserve space. Please refer to Table 4 for their corresponding full dataset names.

| Task | SA | | | | TD | | | | NLI | | | | NER | | | | EQA | | | |
|---|---|---|---|---|---|---|---|---|---|---|---|---|---|---|---|---|---|---|---|---|
| Dataset | AZ | DS | SE | SST | CC | AC | IH | TG | MN | AN | CN | WN | FN | CoNLL | ENER | WNUT | SQuAD | AQA | NQA | SQA |
| Vanilla | 90.94 | 47.38 | 49.90 | 75.16 | 87.15 | 57.47 | 63.77 | 68.83 | 89.40 | 36.19 | 37.06 | 63.32 | 79.89 | 69.10 | 48.01 | 45.45 | 93.14 | 51.19 | 63.77 | 37.47 |
| EDA | 91.66 | 46.39 | 48.02 | 75.82 | 87.15 | 57.47 | 63.77 | 68.83 | 65.57 | 34.50 | 46.25 | 46.40 | - | - | - | - | - | - | - | - |
| FreeLB | 91.39 | 47.94 | 47.88 | 76.66 | 85.63 | 63.55 | 62.22 | 67.98 | 89.83 | 36.13 | 40.94 | 63.58 | 80.08 | 66.66 | 50.84 | 47.77 | 93.51 | 51.07 | 65.03 | 39.57 |
| FL | 91.02 | 46.20 | 50.11 | 76.76 | 87.10 | 57.72 | 62.29 | 67.66 | 89.17 | 36.53 | 39.26 | 63.32 | 79.30 | 61.04 | 50.49 | 45.51 | 92.97 | 50.64 | 63.96 | 36.03 |
| LS | 90.19 | 47.31 | 46.35 | 76.19 | 86.65 | 57.84 | 62.66 | 67.98 | 89.68 | 36.50 | 39.36 | 62.86 | 79.66 | 68.81 | 48.04 | 47.21 | 93.32 | 50.93 | 63.97 | 34.51 |
| ES | 50.72 | 41.83 | 54.98 | 63.36 | 82.99 | 47.02 | 61.05 | 65.32 | 77.67 | 35.16 | 17.79 | 17.79 | - | - | - | - | - | - | - | - |

Table 6: Evaluations of LLMs on BOSS. Small Ref represents the results of supervised fine-tuned small models in Table 5 (Vanilla). We observe that given enough ID data, fine-tuning domain-specific models is predominant when testing on ID examples. In contrast, LLMs with In-context learning should be given priority on OOD instances.

| Task | | SA | | | | TD | | | | NLI | | | | NER | | | | EQA | | | |
|---|---|---|---|---|---|---|---|---|---|---|---|---|---|---|---|---|---|---|---|---|---|
| Dataset | | AZ | DS | SE | SST | CC | AC | IH | TG | MN | AN | CN | WN | FN | CoNLL | ENER | WNUT | SQuAD | AQA | NQA | SQA |
| Small Ref | Full-data | 90.94 | 47.38 | 49.90 | 75.16 | 88.63 | 50.67 | 62.29 | 65.74 | 89.40 | 36.19 | 37.06 | 63.32 | 79.89 | 69.10 | 48.01 | 45.45 | 93.14 | 51.19 | 63.77 | 37.47 |
| T0-3B | 0-shot | 88.33 | 43.80 | 41.08 | 58.76 | 10.60 | 80.92 | 38.48 | 44.15 | 44.50 | 35.00 | 46.29 | 39.82 | 0 | 0 | 0 | 0 | 80.84 | 41.89 | 54.23 | 39.58 |
| T5-3B | 0-shot | 84.55 | 33.63 | 34.27 | 37.68 | 21.03 | 75.94 | 40.72 | 44.04 | 35.44 | 33.53 | 46.29 | 37.16 | 0 | 0 | 0 | 0 | 46.64 | 18.88 | 21.37 | 13.42 |
| | ICL | 84.55 | 33.63 | 34.27 | 37.68 | 21.03 | 75.94 | 40.72 | 44.04 | 35.44 | 33.53 | 46.29 | 37.16 | 0 | 0 | 0 | 0 | 35.50 | 17.16 | 9.13 | 7.75 |
| | 5-shot | 66.73 | 42.73 | 44.12 | 63.92 | 66.08 | 43.01 | 57.58 | 54.47 | 33.19 | 34.94 | 11.24 | 47.60 | 0 | 0 | 0 | 0 | 57.67 | 23.74 | 26.02 | 15.10 |
| | Full-data | 90.63 | 51.11 | 50.93 | 74.13 | 86.32 | 60.75 | 63.12 | 70.64 | 90.64 | 45.97 | 44.57 | 65.16 | 46.80 | 48.28 | 56.63 | 34.97 | 94.38 | 57.48 | 66.40 | 34.75 |
| LLaMA-7B | 0-shot | 75.66 | 54.05 | 37.60 | 46.43 | 67.72 | 43.70 | 57.33 | 59.98 | 32.81 | 26.83 | 68.18 | 44.44 | 0.49 | 0.38 | 0.07 | 0 | 58.98 | 30.22 | 40.78 | 45.80 |
| | ICL | 84.30 | 55.19 | 42.66 | 59.14 | 89.70 | 20.17 | 63.62 | 59.68 | 39.81 | 33.50 | 19.30 | 38.17 | 0.63 | 0.70 | 0.81 | 0.16 | 67.57 | 37.35 | 44.15 | 43.78 |
| | ICL* | - | 52.26 | 47.25 | 53.80 | - | - | - | 60.74 | - | 33.47 | - | 37.98 | - | 0.21 | - | 0 | - | 37.09 | 48.32 | - |
| LLaMA-13B | 0-shot | 81.35 | 56.48 | 42.73 | 59.59 | 89.87 | 19.60 | 62.65 | 58.80 | 32.07 | 24.43 | 47.06 | 38.14 | 0.16 | 1.00 | 0.47 | 0.00 | 66.90 | 37.34 | 45.15 | 55.60 |
| | ICL | 82.72 | 54.71 | 40.63 | 57.69 | 83.91 | 38.88 | 66.96 | 67.87 | 38.58 | 34.28 | 20.42 | 38.26 | 0.37 | 1.11 | 0.85 | 0.00 | 69.71 | 41.90 | 45.32 | 58.67 |
| | ICL* | - | 46.56 | 38.46 | 53.05 | - | - | - | 68.09 | - | 36.00 | - | 37.31 | - | 0.50 | - | 0.00 | - | 40.99 | 49.30 | - |
| Davinci3 | 0-shot | 82.92 | 70.09 | 66.86 | 77.13 | 74.20 | 69.14 | 60.56 | 69.04 | 69.74 | 48.44 | 37.40 | 52.24 | 38.96 | 47.03 | 35.52 | 32.54 | 83.68 | 55.50 | 54.78 | 63.74 |
| | ICL | 84.60 | 74.88 | 68.88 | 77.23 | 78.18 | 65.98 | 61.96 | 77.00 | 71.21 | 50.88 | 55.33 | 53.32 | 48.56 | 57.08 | 46.27 | 42.10 | 82.33 | 57.58 | 54.79 | 69.55 |
| | ICL* | - | 75.74 | 65.14 | 73.95 | - | - | - | 75.29 | - | 49.91 | - | 53.06 | - | 55.83 | - | 40.71 | - | 57.90 | 56.15 | - |
| Turbo | 0-shot | 85.63 | 74.46 | 68.33 | 77.04 | 76.10 | 80.57 | 52.69 | 72.76 | 68.78 | 46.84 | 50.82 | 48.95 | 36.53 | 51.36 | 30.29 | 33.35 | 81.21 | 50.49 | 47.68 | 64.38 |
| | ICL | 87.75 | 77.18 | 65.26 | 75.07 | 60.16 | 80.56 | 54.67 | 76.07 | 68.12 | 49.47 | 49.59 | 49.47 | 40.21 | 57.02 | 38.77 | 35.25 | 82.82 | 50.78 | 50.37 | 64.10 |
| | ICL* | - | 79.10 | 66.79 | 74.95 | - | - | - | 73.90 | - | 49.72 | - | 49.46 | - | 56.12 | - | 36.38 | - | 54.29 | 56.63 | - |

effective approach. Conversely, the inductive-prior-based ensemble is the worst, consistently leading to performance degradation except for the SemEval dataset.

*Takeaway 2: Method effectiveness is consistent, yet limited to specific datasets.* Across multiple datasets, various methods consistently demonstrate (in)effectiveness, as indicated in Table 5. However, no method consistently performs well on all datasets within the same task.

To summarize,**current approaches fall short of meeting the expectations in enhancing the OOD robustness of models**, highlighting the urgent demand for more advanced improvement techniques.

## 4.2 Large Language Models

LLMs are receiving increasing attention from NLP researchers. Considering the impressive zero/few-shot ability of LLMs, and the large difference in their fine-tuning and in-context learning paradigms, it is also intriguing to benchmark their generalizability on various downstream tasks and explore the best paradigm to leverage their power.

**Setting.** We consider three prominent state-of-the-art LLMs, LLaMA-7B and LLaMA-13B (i.e., LLaMA-series) [89] ,OpenAI text-davinci-003 [9] and gpt-3.5-turbo (denoted as Davinci3 and Turbo respectively). For comparison, we include two relatively smaller (yet still large) models, T0-3B [81] and T5-3B. We perform zero-shot inference based on task instructions on all these models since this paradigm is the most general. Then, we adopt other paradigms in a model-specific way. For T5-3B, we include 5-shot and full-data fine-tuning, and we also select examplars from the ID dataset for in-context learning. For the other three LLMs, we apply in-context learning with two kinds of contexts, one from the ID dataset and another from the original training split of the evaluated OOD dataset, denoted as ICL and ICL* respectively. The implementation details are in Appendix E.2.

**Results.** We present the results in Table 6, where the mark '-' means that ICL* paradigm is not applicable for those datasets due to the absence of a training split or the limit of context window size. Our findings can be summarized as follows:

*Takeaway 1: Fine-tuning small domain-specific models is superior when enough training data is available, while LLMs may be favored in low-resource scenarios.* To be specific, supervised fine-tuned small models and T5-3B with the entire dataset consistently achieves the best performance on the ID dataset, especially for the structure prediction task (e.g., NER). In contrast, LLMs exhibit

better performance on most OOD datasets. This observation reinforces the view that large pre-trained models possess strong generalization capabilities, whereas, with sufficient training data, an accurate estimate of data distribution can be achieved even without a large number of parameters [76].

*Takeaway 2: In-context learning always brings no gains to the generalization ability of small models, while it generally helps Turbo and significantly improves LLaMA-series and Davinci3.* For small models like T5-3B, the performance of in-context learning is the same with or even worse than the zero-shot inference. For Turbo, providing ID examples for in-context learning presents advantages on nearly two-thirds of the datasets, with the NER task benefiting the most. For LLaMA-series and Davinci3, the superiority of in-context learning is prominent as it enhances performances on most of the datasets.

*Takeaway 3: Examples from ID datasets are generally more effective for in-context learning than those from the original training split of the testing OOD dataset.* Specifically, when considering samples from our OOD datasets as contexts, the performance of Turbo is comparable to using ID samples, whereas the LLaMA-series and Davinci3 models consistently exhibit inferior performance compared to using ID examples as contexts. However, all models show improved performance on the EQA task when contexts from our OOD datasets are utilized. This may be attributed to the variations in sample length or question styles across EQA datasets, hence models acquire more precise instructions from the original training samples. The overall ineffectiveness of ICL* can be explained by the findings of [69]. According to [69], in-context demonstrations aim to guide the models to learn the target label space, instead of the feature-label mapping. The ID examples contain more diverse information due to the construction process of our benchmark. Thus, ID examples can better prompt the language models to locate the target label space, compared to the OOD examples that may target a specific domain.

**Discussion.** Two paradigms are prevalent in developing downstream NLP systems: leveraging general-purpose LLMs or gathering domain-specific data for fine-tuning smaller models. For the first paradigm, the overarching objective of general-purpose LLM development is to employ a single model for solving various downstream tasks [9]. Consequently, LLMs are anticipated to exhibit high performance on both ID and OOD datasets. However, our study exposes the shortcomings of LLMs on ID datasets when compared to fine-tuned domain-specific models. Considering the higher inference and deployment costs associated with LLMs, substantial progress is still needed to effectively improve LLMs in developing downstream applications, particularly for challenging tasks like EQA. For the second paradigm, our study reveals the limitations of fine-tuning models on ID datasets for OOD performance in comparison to LLMs. Thus, further research is required to develop advanced techniques that enhance the robustness of fine-tuned domain-specific models. Overall, the existing two prevalent paradigms still fall short in addressing the OOD problem in NLP, necessitating further advancements and effective approaches.

However, we also note that there can exist confounders in our evaluations. It is still ambiguous which datasets are indeed OOD to LLMs, given that LLMs have been pre-trained on massive public corpora. The potential data contamination issue can result in overinflated performance on our OOD datasets, tangling the credit of the memory and generalizability of LLMs. The only confirmed distribution shift for LLMs is the temporal shift, necessitating the evaluation based on data released subsequent to their pre-training data collection cut-off. Therefore, the NLP community demands new downstream datasets independent of the pre-training corpus to meet the evaluation requirements for LLMs.

## 5  Related Work

**Distribution shifts in NLP** has been widely studied in various forms. We examine several representative cases as outlined below. Domain shift refers to the challenge of testing data originating from diverse domains, often due to data collection from various sources [63, 40, 53, 79]. Temporal shift examines the degradation of models' performance over time [42, 1]. Spurious correlation examines the issue of models acquiring dataset-specific knowledge on ID data, which may not generalize effectively to OOD data [66, 73, 91, 32, 37, 16, 17]. Additionally, a requirement is for models to exhibit robustness when confronted with artificially constructed OOD samples. One typical type is malicious adversarial attacks, which involve assessing the resilience of models against inputs crafted by malevolent adversaries [56, 51, 112]. These inputs, distinct from ID samples, have the potential to induce model failures [12]. Adversarial attacks can also be effectively utilized to simulate

diverse user inputs to examine models' robustness in the real world [98, 13, 33]. Another category is backdoor attacks, characterized by intentionally introduced spurious correlations that can be exploited by attackers for their advantage [18, 55].

**OOD Evaluation in NLP** can be broadly classified into automatic and static evaluation approaches. Automatic evaluation utilizes diverse textual transformation techniques, such as introducing typos, to conduct a rigorous evaluation of OOD robustness. Three essential elements in the automatic OOD evaluation encompass the establishment of suitable transformation methods, evaluation metrics, and effective techniques to ensure sample validity [34, 98]. Static evaluation, in contrast to automated methods, offers the advantage of constructing benchmarks with higher quality, resulting in an improved estimation of OOD robustness. Numerous OOD benchmarks have been introduced, focusing on adversarial attacks [96] or spurious correlations [117, 66]. A relevant study to ours is GLUE-X [107], which establishes an OOD benchmark derived from the GLUE benchmark [93]. Nevertheless, they do not establish a coherent benchmark construction protocol and primarily rely on dataset selection driven by popularity, incorporating datasets into the benchmark without comprehensive explanation and seemingly opting for a somewhat arbitrary selection, thus lacking a systematic approach.

## 6 Conclusion

We revisit OOD robustness research in NLP, identifying deficiencies in benchmarks and evaluation. Correspondingly, a benchmark construction protocol and an OOD robustness evaluation suite are proposed to facilitate future research. The correlation between OOD and ID performance, the effectiveness of existing methods, and the challenges faced by LLMs are investigated.

## Limitation

We identify two limitations in this work. First, as discussed in section 4.2, due to the lack of new datasets in the community, there is a possibility that some datasets have been included in the pre-training corpus of LLMs, so they may not be suitable to test the generalizability of recent LLMs. However, we note that with our benchmark construction protocol, we can easily update the benchmark as new datasets come out. Second, we only consider five tasks in this benchmark, which is not a comprehensive collection of current NLP literature. We explain the reason for the current task selection in Appendix A.1.

## Acknowledgement

This work is sponsored by the Tsinghua-Toyota Joint Research Fund.

Lifan Yuan and Yangyi Chen initiated the project. Lifan Yuan, Yangyi Chen, and Ganqu Cui designed the experiments. Lifan Yuan, Yangyi Chen, and Hongcheng Gao constructed the AdvCivil dataset. Lifan Yuan conducted experiments and wrote the paper. Yangyi Chen and Ganqu Cui revised the paper. Everyone participated in the discussion. Heng Ji, Zhiyuan Liu, and Maosong Sun advised the project.

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

Table 7: The comparison among different semantic representation methods, from which we can observe that despite different methods varying to some extent, the trend of similarity almost remains unchanged.

| Method | SimCSE | | | | | | | ESimCSE | | | | | | | Sentence-Transformer | | | | | | |
|---|---|---|---|---|---|---|---|---|---|---|---|---|---|---|---|---|---|---|---|---|---|
| Src \| Tgt | AZ | DSC | DS | IMDb | SE | SST | Yelp | AZ | DSC | DS | IMDb | SE | SST | Yelp | AZ | DSC | DS | IMDb | SE | SST | Yelp |
| AZ | 100 | 86.02 | 57.30 | 36.67 | 24.74 | 33.70 | 49.22 | 100 | 95.27 | 88.46 | 76.07 | 76.50 | 76.19 | 81.75 | 100 | 90.25 | 46.57 | 48.62 | 30.65 | 48.92 | 49.23 |
| DSC | 86.02 | 100.00 | 59.15 | 54.55 | 31.70 | 44.40 | 55.45 | 95.27 | 100 | 84.22 | 80.06 | 76.68 | 80.97 | 79.26 | 90.25 | 100 | 37.27 | 61.54 | 25.28 | 54.79 | 49.49 |
| DS | 57.30 | 59.15 | 100 | 32.69 | 28.17 | 19.68 | 88.99 | 88.46 | 84.22 | 100 | 70.64 | 79.28 | 74.41 | 90.52 | 46.57 | 37.27 | 100 | 24.76 | 36.94 | 32.65 | 78.37 |
| IMDb | 36.67 | 54.55 | 32.69 | 100 | 46.95 | 84.62 | 39.88 | 76.07 | 80.06 | 70.64 | 100 | 72.16 | 83.73 | 72.82 | 48.62 | 61.54 | 24.76 | 100 | 25.22 | 84.26 | 34.05 |
| SE | 24.74 | 31.70 | 28.17 | 46.95 | 100 | 40.45 | 24.03 | 76.50 | 76.68 | 79.28 | 72.16 | 100 | 76.75 | 68.64 | 30.65 | 25.28 | 36.94 | 25.22 | 100 | 40.66 | 19.26 |
| SST | 33.70 | 44.40 | 19.68 | 84.62 | 40.45 | 100 | 19.43 | 76.19 | 80.97 | 74.41 | 83.73 | 76.75 | 100 | 62.44 | 48.92 | 54.79 | 32.65 | 84.26 | 40.66 | 100 | 24.58 |
| Yelp | 49.22 | 55.45 | 88.99 | 39.88 | 24.03 | 19.43 | 100 | 81.75 | 79.26 | 90.52 | 72.82 | 68.64 | 62.44 | 100 | 49.23 | 49.49 | 78.37 | 34.05 | 19.26 | 24.58 | 100 |

# Appendix

# A    Frequently Asked Questions

## A.1    What is the rationale for current task selection and why not include more difficult tasks?

The chosen tasks within BOSS evaluate models across natural language understanding, structured data prediction, and question answering, which are core language model competencies. In constructing our benchmark, we did consider extending other tasks such as NLG and commonsense reasoning, but in practice, we found a lot of difficulties.

For NLG tasks, the primary concern lies in evaluation. Since different datasets exhibit distinct text styles, an in-domain (ID) model might generate responses with styles diverging from the reference answers when tested on out-of-domain (OOD) datasets. However, current NLG metrics evaluate predictions based on their resemblance to reference answers, which might not accurately reflect text quality in scenarios with a vast output space [35]. An emerging alternative involves employing Language Models like GPT-4 for scoring predictions. However, this method is cost-intensive and lacks reproducibility due to unpredictable updates. Considering these evaluation challenges, including generation tasks within this OOD benchmark might not be appropriate.

For commonsense reasoning, multiple datasets exist from various sources, each demanding distinct knowledge. These differences are substantial enough that knowledge gained from an ID dataset might not be applicable to OOD datasets. For instance, HellaSwag [114] necessitates basic world knowledge and logical reasoning to complete a sentence, while StepGame [82] relies on spatial imagination without requiring world knowledge. The dissimilarity in required abilities suggests that models should acquire knowledge through pre-training rather than fine-tuning on an ID dataset and then transferring it to OOD tasks. This approach aligns more reasonably with the task's nature.

## A.2    Why is SimCSE chosen to measure distribution shift?

We refer to [105] as a representative study measuring text distribution shift via semantic vector distance. While [105] employs CLIP's encoder for multimodal contexts, our NLP-specific context steers us towards SimCSE for semantic representation.

To the best of our knowledge, SimCSE and sentence-BERT [80] are the most widely used models and SimCSE is more advanced according to the SimCSE paper, hence we choose SimCSE as the semantic representation model. In this discussion period, we also try the other variants of SimCSE, ESimCSE [104], and the best model in the Sentence Transformers leaderboard [6] ('all-mpnet-base-v2'). The results are shown in Table 7. From the results, despite different methods varying to some extent, the trend of similarity almost remains unchanged. Therefore, we consider that the essential is to adopt an advanced semantic representation model, and that which particular model we choose will not lead to differences in our selection.

# B    Survey

Our survey draws from two NLP taxonomy papers, one focusing on robustness [99] and the other on generalization [43]. We meticulously examine all the references cited in these papers, presenting

---

[6] https://www.sbert.net/docs/pretrained_models.html#model-overview

Table 8: Survey of papers targeting to address OOD robustness across the five tasks as this paper. Column "Paper" signifies individual papers sorted by date of publication, column "Tasks" details the evaluated tasks, and column "Datasets" lists the encompassed datasets, with each row representing ID dataset → OOD datasets.

| Paper | Tasks | Datasets |
|---|---|---|
| [46] | SA | IMDb → Amazon, SemEval, Yelp |
| [85] | NLI | MNLI, SNLI, SICK → MNLI, SNLI, SICK |
| [30] | EQA | SQuAD, NewsQA, TtiviaQA, SearchQA, HotpotQA, Natural Questions → BioASQ, DROP, DuoRC, RACE, RelationExtraction, TextbookQA |
| [86] | EQA | SQuAD, NewsQA, SearchQA, TriviaQA, HotpotQA → CQ, CWQ, ComQA, WikiHop, DROP |
| [111] | EQA | SQuAD → TriviaQA, QuAC, QA-SRL, QA-ZRE |
| [40] | SA | SST2, IMDb, Yelp, Amazon → SST2, IMDb, Yelp, Amazon |
| [72] | SA
NLI | Amazon, Yelp, IMDb, SST2 → Amazon, Yelp, IMDb, SST2
MNLI, ANLI → MNLI, ANLI |
| [71] | TD | Wiki → Founta, Waseem |
| [91] | NLI | MNLI → HANS |
| [95] | NLI | MNLI, SNLI, FEVER → ANLI |
| [67] | EQA | SQuAD → Squadshift |
| [103] | SA
NLI | SST-2 → Senti140, SemEval, Amzbook, Yelp, IMDB, IMDb-Cont., IMDb-CAD
SNLI → MNLI, SNLI-CAD, break, DNC, stress, diagnostic |
| [11] | SA
NLI | SST2$ → IMDb-Cont, IMDb-CAD
MNLI → ANLI, HANS |
| [15] | SA
NLI
EQA | SST2 → IMDB
MNLI → ANLI
SQuAD → Adv SQuAD |
| [27] | NLI | FEVER, MNLI → Symmetric, HANS |
| [57] | NLI | MNLI → SNLI, SciTail |
| [52] | EQA | SQuAD → BioASQ, New Wikipedia, New York Times, Reddit posts, Amazon Reviews |
| [97] | SA | SST2, Amazon kitchen, Amazon electronics → SST, Amazon kitchen, Amazon electronics |
| [74] | SA
NLI | SST2 → Senti140, SemEval, Yelp, IMDB, Contrast, CAD
MNLI → PI, LI, HANS, WaNLI, SNLI, ANLI |
| [107] | SA
NLI | SST → IMDB, Yelp, Amazon
MNLI → MNLImm, SNLI, SICK |
| [62] | TD | HateXplain |
| [47] | NLI | MNLI,SNLI → HANS |
| [106] | NER | CoNLL → CrossNER |
| [19] | NER | OntoNotes → I2B2'14, CONLL'03, WNUT'17, GUM, Few-NERD |
| [109] | EQA | SQuAD → NewsQA, SearchQA, TriviaQA, HotpotQA, Natural Questions |

a subset of references that specifically address OOD robustness in sentiment analysis (SA), toxic detection (TD), natural language inference (NLI), name entity recognition (NER), and extractive question answering (EQA) tasks. The summarized findings are outlined in Table 8. Through the survey, we can observe that there is no single uniform set of datasets for evaluation in NLP, namely no "standardized OOD benchmark suites tailored for NLP". And we can find that existing work includes evaluation datasets without particular consideration while mainly directly using popular datasets, i.e. adopting "popularity-based dataset selection strategies".

## C  Datasets

In section 2.4, we outlined the datasets utilized in the BOSS benchmark. In this section, we provide additional descriptions for the datasets not covered in the main body of this paper, and elaborate on the data processing procedures for all candidate datasets.

### C.1  Description

**Sentiment Analysis.**  **DSC** [48] consists of product reviews of 24 different categories in Amazon, with binary sentiments. **IMDb** [64] is another binary classification dataset with movie reviews from the IMDb website. **Yelp** [116] dataset is constructed by extracting product reviews from the Yelp website.

**Toxic Detection.**  **AbuseAnalyzer** [10] collects abusive contents from from Grab, with labels of `Hateful` and `non-Hateful`. **AdvCivil** is a new adversarial dataset for toxic detection introduced in this paper. We employ the adversarial attack technique proposed by [12] to simulate real-world ma-

licious attacks by introducing typos and distracting sentences into the original sentence. Additionally, a post-human inspection process is undertaken to validate the constructed samples. Three human annotators are employed to filter out label-inconsistent and semantically broken samples. **Hate Speech** [21] covers posts on various topics and nationalities from several sub-forums on Stormfront. **HSOL** [20] extracts tweets containing pre-defined toxic words or phrases and categorizes them into three classes: `hate speech`, `offensive but not hate speech`, or `neither offensive nor hate speech`. **OLID** [113] collects tweets and annotates each of them according to whether the text is offensive or not.

**Natural Language Inference.**   **BioNLI** [5] is created from abstracts of biomedical publications on PubMed, with each sample labeled as `entailed` or `not-entailed` and regarding the experimental evidence as the premises. The `entailed` samples directly adopt the conclusions of mechanistic information in the abstracts as the hypothesis, while the `not-entailed` samples manipulate the information to generate pseudo conclusions with nine different strategies. **CB** [22] consists of clause-embedded texts from Wall Street Journal, British National Corpus, and Switchboard, each premise containing an embedded clause and the hypothesis being the extraction of the clause. **DocNLI** [110] processes five existing datasets of different tasks into NLI formats, all from news or Wikipedia. The premises are all at the document level, while the hypotheses have a variety of different granularities. **SNLI** [7] extracts image captions from Flickr30k as premises, and generates hypotheses by human annotation.

**Name Entity Recognition.**   **CrossNER** [61] collect sentences from Wikipedia on topics of artificial intelligence, literature, music, natural science, and politics, covering seven coarse-grained entity types in Few-NERD except for the `Building`.

**Extractive Question Answering.**   **HotpotQA** [108] is based on Wikipedia, with each question requiring information from two articles to answer. **NaturalQuestions** [50] takes real queries from Google search engine as questions, corresponding search results as contexts, and human-annotated texts as answers. **Trivia-QA** [45] collects long documents from Wikipedia and constructs questions that can not be directly answered by span extraction. **SQuAD Shifts** [67] inherits the dataset construction pipeline from SQuAD, but collects passages from more diverse sources, including Wikipedia, New York Times, Reddit, and Amazon.

## C.2   Processing

### C.2.1   Sentiment Analysis

Generally, datasets for this task are label-imbalanced, which is detrimental to model performance. Therefore, If not specified, we tackle this issue by aligning $n_i$ with $\min\{n_1, \ldots, n_{|classes|}\}$ by discarding redundant samples in each class, where $|classes|$ is the number of classes and $n_i$ is the number of samples for label $i$ in each dataset, $i \in \{1, \ldots, |classes|\}$.

**Amazon** has 29 subsets of different products. To encourage diversity of the review texts, we aim to merge all the subsets together. However, the scale of the entire dataset is prodigious and the sizes of the subsets vary greatly, making model training time-consuming and potentially ignoring the effect of certain types of reviews. Therefore, we first reduce large subsets to 20k samples, and then drop samples of stars 2 or 4 and maintain samples of starts 1, 3, and 5. After that, we get a ternary classification of the entire dataset, split into training and test datasets by 9:1. Finally, for label balance, we sample 10k reviews for each class in the training dataset and discard the rest.

**DSC** is a binary classification dataset, thus samples of both labels will be retained. The final training and test dataset are formed by combining all types of product reviews.

**Dynasent** has two rounds of training and test datasets, which are incorporated together to constitute a unified training and test dataset.

**IMDb** and **SemEval** only require label balancing processing.

**SST** samples are annotated with a float score indicating the sentiment, ranging from 0 to 1. We follow the common practice of SST-5 to equally divide the score into 5 bins, i.e. mapping score $0\tilde{0}.2$ to

label 1, and so on. Then we drop samples of labels 1 and 3, similar to Amazon, to adapt for ternary classification.

**Yelp** reviews are also rated from 1 to 5 and retained only samples of stars 1, 3, and 5 following Amazon and SST. Due to the large amount of data, we sample 10k reviews from the training dataset for each label.

### C.2.2    Toxic Detection

Datasets for this task are collected from online forums and social media, therefore the texts are commonly dirty, containing some meaningless strings such as `@username` in the beginning. We clean the texts by removing `@username`, emoji, tags (with a symbol #), and URLs. Moreover, the label balancing processing is the same as for Sentiment Analysis datasets.

**AbuseAnalyzer** does not have a train/test split, thus we divide the original dataset into a training dataset and a test dataset by 8:2.

**Civil Comments** samples with the toxicity score $\geq 0.5$ are considered toxic samples while others whose toxicity score $< 0.5$ are in the benign class.

**Hate Speech** is divided into training and test datasets by 8:2.

**HSOL** has three types of label: `hate speech`, `offensive but not hate speech`, or `neither offensive nor hate speech`. We adapt it to binary classification, treating samples with the first two toxic and samples with the label `neither offensive nor hate speech` as benign.

**Implicit Hate** is annotated into three classes, i.e., `not hate`, `implicit hate` and `explicit hate`, with `implicit hate` and `explicit hate` being considered toxic. In probing experiments, we split the training and test dataset by 8:2. After selecting the Implicit Hate dataset into our benchmark, we treat the entire dataset as the test dataset.

**AdvCivil**, **OLID** and **ToxiGen** do not require specific processing.

### C.2.3    Natural Language Inference

Datasets for NLI do not have a serious label-imbalance problem, hence no corresponding processing is applied.

**ANLI** has three rounds in total, we merge them together to form the final training and test datasets respectively.

**DocNLI** contains some ANLI examples, so we filter those samples out to avoid repeating ANLI.

Other datasets, i.e., **BioNLI**, **CB**, **ContractNLI**, **MNLI**, **SNLI**, and **WANLI**, do not require any other specific processing. But note that the test dataset of MNLI is the `matched` validation dataset, and we do not use the `unmatched` one since the two subsets are too similar.

### C.2.4    Name Entity Recognition

Datasets for NER usually have different name entity tags and thus require label mapping for alignment. For the cross-evaluation experiment, we align the labels of each dataset to CoNLL; while for other experiments, we take the scheme of Few-NERD as the standard. Also, we adapt the datasets to the prevalent BIO schema, which indicates whether a token is the Beginning, the Inside, or the Outside of an entity.

**CoNLL** contains the four most common entity types which are covered in Few-NERD, thus no label mapping is needed.

**CrossNER** constitutes various entity types from diverse domains. Therefore, we design the following mapping for label alignment:

```
label_mapping = {
    "academicjournal": "product",
    "album": "product",
    "algorithm": "miscellaneous",
```

```
    "astronomicalobject": "miscellaneous",
    "award": "miscellaneous",
    "band": "organization",
    "book": "art",
    "chemicalcompound": "miscellaneous",
    "chemicalelement": "miscellaneous",
    "conference": "event",
    "country": "location",
    "discipline": "miscellaneous",
    "election": "event",
    "enzyme": "miscellaneous",
    "event": "event",
    "field": "miscellaneous",
    "literarygenre": "art",
    "location": "location",
    "magazine": "product",
    "metrics": "miscellaneous",
    "misc": "miscellaneous",
    "musicalartist": "person",
    "musicalinstrument": "product",
    "musicgenre": "art",
    "organisation": "organization",
    "person": "person",
    "poem": "art",
    "politicalparty": "organization",
    "politician": "person",
    "product": "product",
    "programlang": "miscellaneous",
    "protein": "miscellaneous",
    "researcher": "person",
    "scientist": "person",
    "song": "art",
    "task": "miscellaneous",
    "theory": "miscellaneous",
    "university": "organization",
    "writer": "person"
}
```

**Few-NERD** requires processing to adapt for the scheme of CoNLL in the probing experiments. Since it has covered all the entity types annotated in CoNLL, the only process is to set other tags, i.e. `building`, `art`, `product`, and `event`, to be `miscellaneous`. Note that this process is also required for CrossNER and WNUT in the cross-evaluation experiment, since these two datasets have already been aligned with Few-NERD.

**WNUT** conduct the following operation to align labels with Few-NERD:

```
label_mapping = {
    "corporation": "organization",
    "creative-work": "art",
    "group": "organization",
    "location": "location",
    "person": "person",
    "product": "product"
}
```

### C.2.5 Extractive Question Answering

Datasets are all normalized to a unified extractive setting with the same format as SQuAD, following MRQA [30].

**AdvQA** has several subsets generated by fooling different models. We adopt the combination of all these subsets for our experiments.

Table 9: Statistics of toxic detection candidate datasets.

| Dataset | Source | # Classes | # Samples Train | Test | Avg. Length Train | Test |
|---------|--------|-----------|-------|------|-------|------|
| AbuseAnalyzer | Grab | 2 | 6,080 | 1,520 | 14.24 | 14.38 |
| AdvCivil | Adversarial | 2 | - | 823 | - | 70.73 |
| Civil Comments | Civil Comments | 2 | 60,000 | 97,320 | 50.09 | 51.15 |
| Hate Speech | Stormfront | 2 | 8,562 | 2,141 | 18.07 | 18.1 |
| HSOL | Twitter | 2 | 5,823 | 2,485 | 13.37 | 13.1 |
| Implicit Hate | Twitter | 2 | - | 21,480 | - | 16.81 |
| OLID | Twitter | 2 | 13,240 | 860 | 19.62 | 23.16 |
| ToxiGen | Synthetic | 2 | 8,960 | 940 | 18.14 | 18.63 |

Table 10: SimCSE scores between each pair of datasets regarding the toxic detection task. AA: AbuseAnalyzer; AC: AdvCivil; CC: Civil Comments; HS: Hate Speech; IH: Implicit Hate; TG: ToxiGen.

| Train \| Test | AA | AC | CC | HS | HSOL | IH | OLID | TG |
|---------------|-----|-----|-----|-----|------|-----|------|-----|
| AbuseAnalyzer | **100** | 80.05 | 79.60 | 78.89 | 73.97 | 75.78 | 85.33 | 63.06 |
| AdvCivil | 80.05 | **100** | 95.17 | 59.07 | 51.35 | 69.54 | 74.49 | 58.63 |
| Civil Comments | 79.60 | 95.17 | **100** | 66.89 | 50.96 | 69.05 | 76.20 | 62.09 |
| Hate Speech | 78.89 | 59.07 | 66.89 | **100** | 64.96 | 76.02 | 70.03 | 74.00 |
| HSOL | 73.97 | 51.35 | 50.96 | 64.96 | **100** | 41.73 | 69.80 | 43.32 |
| Implicit Hate | 75.78 | 69.54 | 69.05 | 76.02 | 41.73 | **100** | 64.60 | 79.79 |
| OLID | 85.33 | 74.49 | 76.20 | 70.03 | 69.80 | 64.60 | **100** | 51.76 |
| ToxiGen | 63.06 | 58.63 | 62.09 | 74.00 | 43.32 | 79.79 | 51.76 | **100** |

Table 11: The OOD performance of the T5-large when trained on the Civil Comments dataset. AA: AbuseAnalyzer; AC: AdvCivil; CC: Civil Comments; HS: Hate Speech; IH: Implicit Hate; TG: ToxiGen.

| Train \| Test | AA | AC | CC | HS | HSOL | IH | OLID | TG |
|---------------|-----|-----|-----|-----|------|-----|------|-----|
| Civil Comments | 74.41 | 57.47 | **87.15** | 80.43 | 75.98 | 63.77 | 85.00 | 68.83 |

**SQuAD** can be used in the original format, while other datasets i.e. **HotpotQA**, **NaturalQuestions**, **NewsQA**, **SearchQA**, and **Trivia-QA**, are adopted from MRQA to fit the extractive setting.

## D  Dataset Selection for Other Tasks

In section 2.3, we have taken the task of sentiment analysis as an example to demonstrate how to select ID and OOD datasets for our benchmark. Next, we will explain how we choose datasets for other tasks.

### D.1  Toxic Detection

**Candidate Datasets.**  We use the same approach for searching toxic detection datasets as we do for sentiment analysis. We gather several dataset candidates, including Civil Comments, Hate Speech, HSOL, Implicit Hate, OLID, and ToxiGen. We aim to adopt an adversarial dataset for each NLU task, which is lacking in the current literature. Hence, we later construct a new dataset through adversarial attacks on the chosen ID dataset.

**Probing Experiments.**  The setup is the same as sentiment analysis.

**Results.**  The dataset information can be found in Tables 9. The sources are very diverse, with each source, except Twitter, corresponding to only one dataset. For the ID dataset, Civil Comments is significantly larger than all the other datasets, and it is the only one containing more than 10k samples for each class. In addition, Civil Comments contain 6 subtypes of toxicity and 24 types of targeted identity, meeting the requirement for dataset diversity. Considering the dataset size and the text source diversity, we choose Civil Comments as our ID dataset for toxic detection.

Table 12: Statistics of natural language inference candidate datasets.

| Dataset | Source | # Classes | # Samples | | Avg. Length (p) | | Avg. Length (h) | |
|---|---|---|---|---|---|---|---|---|
| | | | Train | Test | Train | Test | Train | Test |
| ANLI | Adversarial | 3 | 162,865 | 3,200 | 54.13 | 54.42 | 9.6 | 10.22 |
| BioNLI | Biomedical | 2 | 5,544 | 68,243 | 215.93 | 217.93 | 26.91 | 29.65 |
| CB | Wall Street Journal & British National Corpus & Switchboard | 3 | - | 306 | - | 56.4 | - | 7.55 |
| ContractNLI | Legal | 3 | 7,191 | 2,091 | 1673.63 | 1708.82 | 12.82 | 12.82 |
| DocNLI | News & Wikipedia | 2 | 942,314 | 263,886 | 318.89 | 385.13 | 47.5 | 71.88 |
| MNLI | Open American National Corpus | 3 | 392,662 | 9,815 | 19.81 | 19.27 | 9.97 | 9.92 |
| SNLI | Flickr | 3 | 549,361 | 9,824 | 12.85 | 13.91 | 7.42 | 7.48 |
| WANLI | Synthetic | 3 | 102,884 | 5,000 | 17.49 | 17.48 | 9.93 | 9.83 |

Table 13: SimCSE scores between each pair of datasets regarding the natural language inference task.

| Train \| Test | ANLI | BioNLI | CB | ContractNLI | DocNLI | MNLI | SNLI | WANLI |
|---|---|---|---|---|---|---|---|---|
| ANLI | **100** | 31.32 | 29.37 | 21.99 | 53.79 | 16.27 | 21.99 | 22.41 |
| BioNLI | 31.32 | **100** | 5.51 | 10.39 | 28.07 | 2.28 | -4.78 | -0.12 |
| CB | 29.37 | 5.51 | **100** | 17.81 | 40.95 | 68.11 | 41.88 | 59.17 |
| ContractNLI | 21.99 | 10.39 | 17.81 | **100** | 19.96 | 3.62 | -7.17 | 6.12 |
| DocNLI | 53.79 | 28.07 | 40.95 | 19.96 | **100** | 34.91 | 21.23 | 28.00 |
| MNLI | 16.27 | 2.28 | 68.11 | 3.62 | 34.91 | **100** | 47.42 | 85.23 |
| SNLI | 21.99 | -4.78 | 41.88 | -7.17 | 21.23 | 47.42 | **100** | 46.80 |
| WANLI | 22.41 | -0.12 | 59.17 | 6.12 | 28.00 | 85.23 | 46.80 | **100** |

Table 14: The OOD performance of the T5-large when trained on the MNLI dataset.

| Train \| Test | ANLI | BioNLI | CB | ContractNLI | DocNLI | MNLI | SNLI | WANLI |
|---|---|---|---|---|---|---|---|---|
| MNLI | 36.19 | 77.63 | 63.40 | 37.06 | 75.67 | **89.40** | 87.32 | 63.32 |

Next, we consider OOD datasets from other text sources. As aforementioned, we first supplement an adversarial dataset for toxic detection, i.e. AdvCivil, based on Civil Comments. The construction details can be found in Appendix C.1. Then we consider the semantic similarity among the dataset candidates. From the results in Table 10, we can filter out AbuseAnalyzer and OLID because of their high similarity with Civil Comments. We also observe a high similarity between AdvCivil and Civil Comments, but we do not disregard the former dataset at this stage since the high similarity has an acceptable attribution considering the construction process. Moreover, both HSOL and Implicit Hate come from tweets, thus they can not coexist in the benchmark. We are leaning to select Implicit Hate since it contains more challenging implicit toxicity.

Thereafter, we have four datasets left, i.e. AdvCivil, Hate Speech, Implicit Hate, and ToxiGen. We still need to drop one dataset based on performance degradation. According to results in Table 11, AdvCivil, Implicit Hate, and ToxiGen can provoke a performance drop of over 20 points. By contrast, the ID model can still achieve accuracy over 80 on Hate Speech, indicating that this shift may be the least challenging. Therefore, we discard Hate Speech and adopt the other three as the OOD datasets. It is also worth noting that Implicit Hate leads to a much more severe performance drop than HSOL, supporting our previous claim. Finally, the distribution shift for toxic detection is Civil Comments → (AdvCivil, Implicit Hate, ToxiGen).

## D.2 Natural Language Inference

**Candidate Datasets.** We investigate datasets on `Paperswithcode` and finally include ANLI, BioNLI, CB, ContractNLI, DocNLI, MNLI, SNLI, and WANLI as the candidates.

**Probing Experiments.** For semantic similarity evaluation, we only feed premises in each dataset to the unsupervised SimCSE model[7], instead of concatenating the premises and hypotheses together. Note that we do not adopt the supervised SimCSE model here, because it was contrastively trained

---

[7] https://huggingface.co/princeton-nlp/unsup-simcse-roberta-large

Table 15: Statistics of named entity recognition candidate datasets.

| Dataset | Source | # Classes | # Samples | | Avg. Length | |
|---|---|---|---|---|---|---|
| | | | Train | Test | Train | Test |
| CoNLL | Reuters | 4 | 14,042 | 3,454 | 14.50 | 13.44 |
| CrossNER | Wikipedia | 7 | 701 | 2,507 | 38.46 | 38.22 |
| E-NER | Legal | 4 | - | 11,692 | - | 34.52 |
| Few-NERD | Wikipedia | 8 | 131,768 | 37,649 | 24.49 | 24.47 |
| WNUT | YouTube & Twitter & StackExchange & Reddit | 6 | 3,395 | 1,288 | 18.48 | 18.16 |

Table 16: SimCSE scores between each pair of datasets regarding the name entity recognition task.

| Train \| Test | CoNLL | CrossNER | E-NER | Few-NERD | WNUT |
|---|---|---|---|---|---|
| CoNLL | **100** | 62.86 | 43.48 | 73.76 | 45.60 |
| CrossNER | 62.86 | **100** | 37.19 | 87.47 | 44.84 |
| E-NER | 43.48 | 37.19 | **100** | 42.29 | 25.85 |
| Few-NERD | 73.76 | 87.47 | 42.29 | **100** | 45.74 |
| WNUT | 45.60 | 44.84 | 25.85 | 45.74 | **100** |

on NLI datasets, thus may lead to a bias in evaluating the semantic similarity between NLI datasets. For cross-evaluation experiments, the setup is the same as semantic analysis.

**Results.**   Dataset information is shown in Tables 12. These NLI datasets are commonly large and their text sources do not overlap. Based on these results, we start to pick the ID dataset. First of all, BioNLI and DocNLI should be excluded from the discussion of ID dataset selection, because they only have two classes but all the other candidates have three categories. Otherwise, the trained ID model may fail to distinguish the class `Neutral` and `Contradiction`, namely suffering from a label shift [43]. Then, consider the dataset size. The majority of candidates contain over 100k training samples, while CB only has about 300 samples in total and ContractNLI has less than 10k samples, much smaller than others. Therefore, these two datasets should be excluded too. Among the remaining datasets, ANLI and WANLI mainly contain adversarial and challenging features, instead of general ones; MNLI claims to be more diverse than SNLI since it consists of both written and spoken English and contains 10 sub-genres, thus more abundant in text styles and topics. Hence, we pick MNLI as the ID dataset.

For OOD datasets, since there is no dataset drawn from the same source with MNLI, only semantic similarity and performance drop should be involved. From Table 13, the similarities between NLI datasets and MNLI are all relatively low, except for WANLI, CB, and SNLI. We will ignore CB and SNLI in the later procedure but still remain WANLI because WANLI is based on MNLI and thus it is reasonable to be somehow similar to MNLI. Thereafter, we examine the performance drop caused by each dataset. As shown in Table 14, the ID model presents little performance degradation on BioNLI and DocNLI, while suffering significant degradation on the other three datasets. Hence, the distribution shifts for NLI will be MNLI → (ANLI, ContractNLI, WANLI).

### D.3   Name Entity Recognition

**Candidate Datasets.**   Since NER datasets typically have different sets of entity type labels that require specific domain knowledge, we loosen the search standard to include the datasets that have partially overlapping entity type labels, rather than requiring an exact match. We find five suitable candidate datasets, i.e. CoNLL, CrossNER, E-NER, Few-NERD, and WNUT. To align their label sets, we process them to be consistent with Few-NERD since we consider the label set of Few-NERD to be the most general.

**Probing Experiments.**   The setup for semantic similarity evaluation follows the way we do for sentiment analysis, but the backbone model used in the performance evaluation here is a DeBERTa-base model, instead of a T5-base. The reason is that T5 requires inputs to be organized with prompts while standard prompt-based tuning methods for NER are still lacking. Hence, T5 does not perform well on NER task, so we resort to fine-tuning the encoder-only model, DeBERTa, as the alternative.

Table 17: The OOD performance of the DeBERTa-large when trained on the Few-NERD dataset.

| Train | Test | CoNLL | CrossNER | E-NER | Few-NERD | WNUT |
|-------|------|-------|----------|-------|----------|------|
| Few-NERD | | 69.10 | 66.63 | 48.01 | **79.89** | 45.45 |

Table 18: Statistics of extractive question answering candidate datasets. AQA: AdvQA; HQA: HotpotQA; NQ: NaturalQuestion; SQA: SearchQA; SS: SQuAD Shifts; TQA: Trivia-QA.

| Dataset | Source | # Samples | | Avg. Length | |
|---------|--------|-----------|------|-------------|------|
| | | Train | Test | Train | Test |
| AdvQA | Adversarial | 30,000 | 3,000 | 117.36 | 114.83 |
| HotpotQA | Wikipedia | 72,928 | 5,901 | 153.56 | 193.21 |
| NaturalQuestion | Google search | 104,071 | 128,36 | 152.03 | 158.8 |
| NewsQA | CNN News | 74,160 | 42,12 | 495.8 | 492.59 |
| SearchQA | Google search & J!Archive | 117,384 | 16,980 | 646.87 | 642.87 |
| SQuAD | Wikipedia | 87,599 | 10,570 | 119.76 | 123.95 |
| SQuAD Shifts | Wikipedia & Amazon & Reddit & New York Times | - | 29,753 | - | 139.1 |
| Trivia-QA | Wikipedia | 61,688 | 7,785 | 673.75 | 672.64 |

Table 19: SimCSE scores between each pair of datasets regarding the extractive question answering task. AQA: AdvQA; HQA: HotpotQA; NQ: NaturalQuestion; SQA: SearchQA; SS: SQuAD Shifts; TQA: Trivia-QA.

| Train | Test | AQA | HQA | NQ | NQA | SQA | SQuAD | SS | TQA |
|-------|------|-----|-----|-----|-----|-----|-------|-----|-----|
| AdvQA | | **100** | 66.85 | 67.66 | 46.74 | 62.72 | 96.29 | 43.05 | 69.62 |
| HotpotQA | | 66.85 | **100** | 81.25 | 65.53 | 83.99 | 72.99 | 39.89 | 87.81 |
| NaturalQuestions | | 67.66 | 81.25 | **100** | 54.67 | 79.26 | 73.5 | 37.06 | 80.67 |
| NewsQA | | 46.74 | 65.53 | 54.67 | **100** | 66.94 | 51.98 | 35.94 | 64.14 |
| SearchQA | | 62.72 | 83.99 | 79.26 | 66.94 | **100** | 69.22 | 48.19 | 94.17 |
| SQuAD | | 96.29 | 72.99 | 73.5 | 51.98 | 69.22 | **100** | 42.83 | 75.72 |
| SQuAD Shifts | | 43.05 | 39.89 | 37.06 | 35.94 | 48.19 | 42.83 | **100** | 48.12 |
| Trivia-QA | | 69.62 | 87.81 | 80.67 | 64.14 | 94.17 | 75.72 | 48.12 | **100** |

Table 20: The OOD performance of the T5-large when trained on the SQuAD dataset.

| Train | Test | AQA | HQA | NQ | NQA | SQA | SQuAD | SS | TQA |
|-------|------|-----|-----|-----|-----|-----|-------|-----|-----|
| SQuAD | | 51.19 | 74.08 | 64.54 | 63.77 | 37.47 | **93.14** | 89.02 | 73.15 |

**Results.** The dataset statistics are shown in Table 15. Few-NERD is much larger than other datasets and contains the largest variety of samples, with 8 coarse-grained and 66 fine-grained types. Therefore, we select it as the ID dataset for the name entity recognition task.

For OOD datasets, since CrossNER shares the same source with the ID dataset Few-NERD, it should not be adopted to our benchmark. The results in Tables 16 and 17 supports this claim from the perspectives of semantic similarity and performance drop, respectively. Then, the remaining three datasets will be used to construct the distribution shifts, i.e., Few-NERD → (CoNLL, E-NER, WNUT).

## D.4 Extractive Question Answering

**Candidate Datasets.** We consider AdvQA, HotpotQA, NaturalQuestions, NewsQA, SearchQA, SQuAD, SQuAD Shifts, and Trivia-QA as the candidates.

**Probing Experiments.** In the experiment of semantics similarity, we take the passages as inputs for the supervised SimCSE model. In the experiment of performance evaluation, we choose T5-base as the backbone model and measure F1 scores.

**Results.** Table 18 shows the dataset statistics. Except for SQuAD Shifts, which does not have a training split, all the other datasets are large in size. However, considering the diversity, we assume that Wikipedia provides the most extensive knowledge base. Hence, we restrict the scope of the ID dataset to those created from Wikipedia. Among the three datasets, HotpotQA, SQuAD, and Trivia-QA, SQuAD is the largest one. and therefore, we adopt SQuAD as the ID dataset.

Then, since SQuAD has been selected to be the ID dataset, HotpotQA and Trivia-QA are considered in the OOD dataset selection. And besides these two datasets, NaturalQuestions also have a high semantic similarity with SQuAD, as shown in Table 19. Therefore, for the distinction of the distributions, NaturalQuestions will not be taken into account either. In contrast, despite a high semantic similarity with SQuAD, we do not exclude AdvQA for this reason. This is because AdvQA is constructed by human adversaries based on SQuAD, aiming to reveal the weakness of models trained on SQuAD. Therefore, we consider the high similarity between AdvQA and SQuAD reasonable and should not disregard AdvQA for this sake. Next, we evaluate the ID model on each candidate dataset to examine the challenge of each distribution shift. From Tabel 20, we can observe that the ID model performs poorly on AdvQA, NewsQA, and SearchQA while maintaining a high F1 score on SQuAD Shifts, indicating that the shift to SQuAD Shifts is the least challenging. Eventually, we establish the distribution shifts for extractive question answering as SQuAD → (AdvQA, NewsQA, SearchQA).

# E    Additional Descriptions and Results of the Analysis and LLMs Evaluation

## E.1    Correlation between ID and OOD performance

### E.1.1    Experimental Setting

To measure the correlation between ID and OOD performance, we train models with various ID performances by manipulating model scales, training steps, available training samples, and tunable parameters following [14]. The details are introduced below.

**Model scale.** We adopt the small, base, and large versions of the T5 and DeBERTa models.

**Available training samples.** For Sentiment Analysis, Toxic Detection, and Natural Language Inference, we randomly sample K samples for each class. For Name Entity Recognition, we adopt the N-way K-shot sampling strategy proposed by [26], ensuring that entities of all N types occur K∼2K times in the sampled dataset. For Extractive Question Answering, we randomly sample K questions. In all experiments, we run five repeated experiments for each K.

**Number of tunable parameters.** We employ parameter-efficient tuning methods to regulate the tuned parameter for adapting to downstream tasks We prioritize adopting the strong approach Adapter [41]. However, in experiments, we find that Extractive Question Answering models can achieve a high ID performance even with a very insignificant amount of parameters being tuned, thus presenting no performance change when the number of tunable parameters increases. Therefore, we turn to an alternative, Soft-prompt [54], for a clearer observation.

**Training steps.** We evaluate model performance at epoch $\{0, 0.1, \ldots, 0.9\} \cup \{1.0, \ldots, \texttt{EPOCH}\}$, where `EPOCH` is the total number of training epochs.

### E.1.2    Full Results

We present the complete results of ID-OOD correlation in Figure 4. Note that due to the variance of data points, we manually check and remove outlier points to make the fitting results clearer. Next, we will illustrate the results task by task.

**Sentiment Analysis.** All figures of SA comprise two straight lines. The smoother line, which mainly fits the results of T5-small models with poor ID performances, stretches from around ID accuracy = 60. Meanwhile, the steeper line accommodates T5-base and T5-large model results and begins around ID accuracy = 85. Despite the different slopes, both lines represent a linear positive correlation between ID performance and OOD generalization, thus we still categorize this kind of relation as Type I.

**Toxic Detection.** This task encompasses two types of ID-OOD relations. AdvCivil stands as an outlier as aforementioned, whereas Implicit Hate and ToxiGen fall under the category of Type I relation, characterized by a straight line that intersects the diagonal line $y = x$ with a flatter slope. This trend indicates that although achieving initially higher performance on OOD examples than on ID examples, the model's performance gap between ID and OOD decreases and eventually reverses as training proceeds

**Natural Language Inference.** All relations are classified as Type III, a non-monotonic correlation. Each graph has a V-shaped curve, consisting of two straight lines with different slopes. Specifically, for ANLI, the first straight line has a shallower slope than the second one, while ContractNLI and WANLI exhibit the opposite trend.

**Named Entity Recognition.** The ID-OOD relations in this task are all typical Type I, with all fitted lines lying below the diagonal line. The CoNLL dataset exhibits the highest correlation to a perfectly linear relationship ($y = x$), while the WNUT dataset has the lowest correlation. This phenomenon indicates that improving OOD performance on CoNLL is the easiest while on WNUT is the hardest.

**Extractive Question Answering.** Type II relation is observed in this task, with varying degrees of curve changes. Specifically, in the SearchQA dataset, the angle of the curve change is the most pronounced, while in the NewsQA dataset it is the smoothest.

## E.2 LLMs Evaluation

**Experiment Setting.** The templates and instructions we used are shown in Tables 21 and 22. We take Dynasent as an example to demonstrate the in-context learning settings for LLMs. The in-context (ID) setting includes examplars drawn from the ID training dataset of the sentiment analysis task, i.e. training dataset of Amazon. However, the in-context (OOD) setting samples the in-context examplars from the training split of the OOD dataset, i.e. training dataset of Dynasent. Note, however, that the latter paradigm may not be applicable to all datasets. All ID datasets are excluded, as well as OOD datasets without a training split. Besides, those datasets whose text length is too long on average would not be taken into account either, as the contexts could exceed the context window of LLMs.

Table 21: Templates for T0-3B and T5-series models.

| Task | Template | Verbalizer |
|------|----------|------------|
| SA | {{input}} All in all , the sentiment was {"mask"} . | negative / positive / neutral |
| TD | {{input}} All in all , the toxicity was {"mask"} . | benign / toxic |
| NLI | Given the two sentences: 
 (1) {{input_1}}. 
 (2) {{input_2}}. 
 Does the first sentence entail the second ? {"mask"} . | Yes / Maybe / No |
| NER | ner text: {{input}} label: {"mask"} | - |
| EQA | Extract the answer to the question from the following context. 
 Question: {{input_1}} 
 Context: {{input_2}}‖ 
 Answer: {"mask"} | - |

Table 22: Instructions for LLMs, including LLaMA-7B, Davinci3, and Turbo.

| Task | Instruction |
|------|-------------|
| SA | ### Instruction ### 
 Solve the sentiment analysis task. Options for sentiment: negative, positive, neutral. 
 ### Format ### 
 Text: {{Text}} // Prediction: {{Prediction}} 
 ### Input ### 
 Text: {{input}} // Prediction: |
| TD | ### Instruction ### 
 Solve the toxic detection task. Options for toxicity: benign, toxic. 
 ### Format ### 
 Text: {{Text}} // Prediction: {{Prediction}} 
 ### Input ### 
 Text: {{input}} // Prediction: |
| NLI | ### Instruction ### 
 Solve the NLI task. Options for entailment relationship: entailment, neutral, contradiction. 
 ### Format ### 
 Premise: {{Premise}} // Hypothesis: {{Hypothesis}} // Prediction: {{Prediction}} 
 ### Input ### 
 Premise: {{input_1}} // Hypothesis: {{input_2}} // Prediction: |
| NER | ### Instruction ### 
 Solve the NER task, identifying the Organization, Person, Location, Miscellaneous, Building, Art, Product, and Event entities from given text. 
 ### Format ### 
 Text: {{Text}} // Entity: Organization: None ‖ Person: Word1 ‖ Location: Word6, Word7 ‖ Miscellaneous: None ‖ Building: None ‖ Art: Word 3 ‖ Product: None ‖ Event: None. 
 ### Input ### 
 Text: {{input}} // Entity: |
| EQA | ### Instruction ### 
 Solve the extractive question answering task. Refering to the passage below and extract answer for the question. The answer should be the shortest phrase as it can be. 
 ### Format ### 
 Passage: {{Passage}} // Question: {{Question}} // Answer: {{Answer}}. 
 ### Input ### 
 Passage: {{input_1}} // Question: {{input_2}} // Answer: |

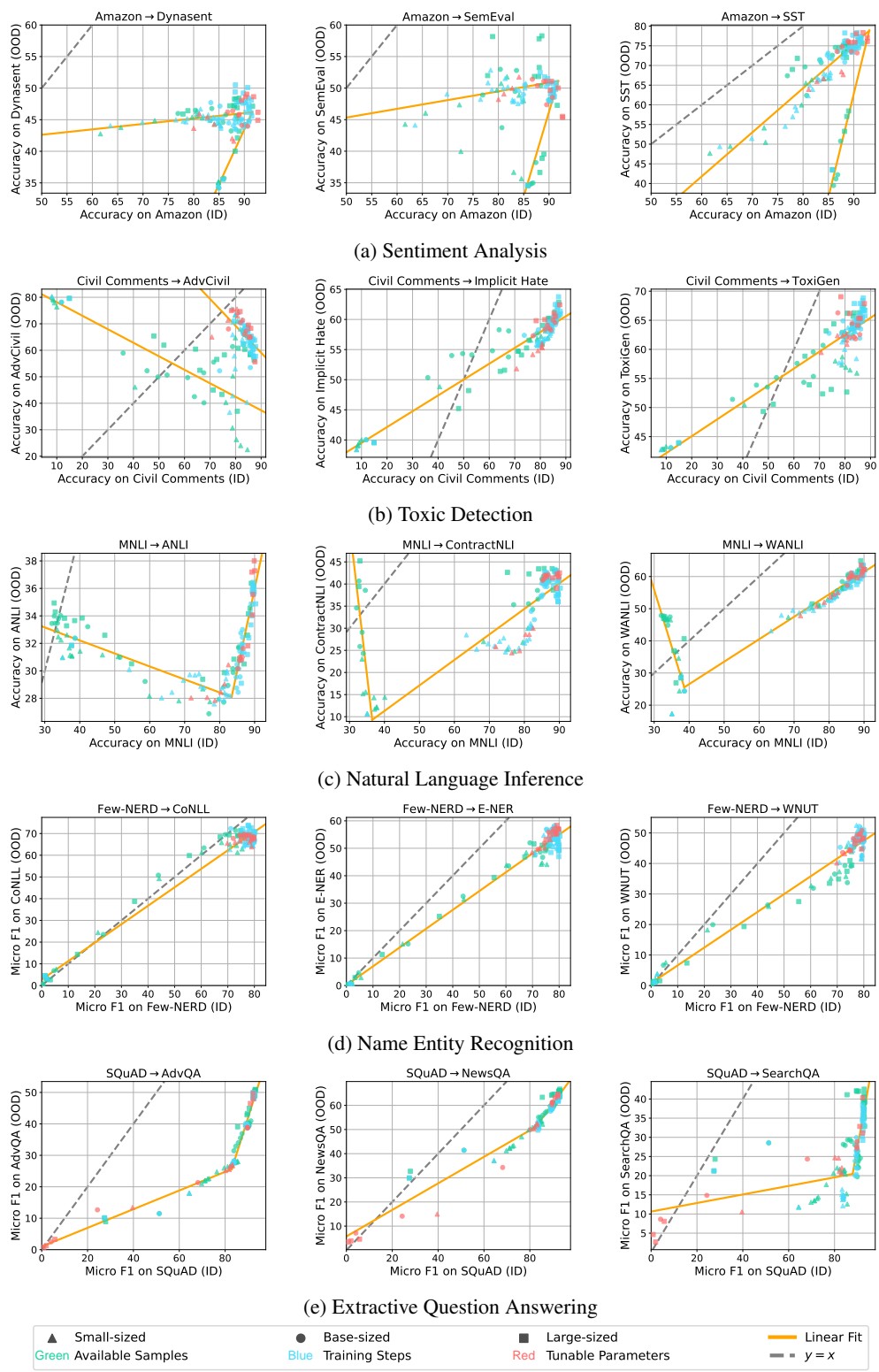

Figure 4: Complete results of the relation between ID and OOD performance.

