# OpenReview forum: "Revisiting Out-of-distribution Robustness in NLP: Benchmarks, Analysis, and LLMs Evaluations"
_NeurIPS.cc/2023/Track/Datasets_and_Benchmarks — NeurIPS 2023 Datasets and Benchmarks Poster_

### Official Review · Reviewer_6XNi · 2023-07-20
**A good benchmark for evaluating the OOD robustness of NLP models, especially LLMs**

**Rating:** 7
**Confidence:** 4

**Strengths:**

- Novelty: This work is trying to address and clarify an important problem in NLP, OOD Robustness. It helps researchers to better understand and handle OOD data in NLP tasks.
- Comprehensive Evaluation: The study evaluates various SOTA methods and models on five diverse NLP tasks. Through intensive experiments and comparison, this work shows a better understanding of their performance and draws fascinating and equally important conclusions.
- Practicality: This study provides valuable guidance for future research through probing experiments, summarizing patterns and detailed analysis. This study’s open source repository will empower researchers to effectively tackle and resolve OOD problems in the future.

**Additional Feedback:**

Please refer to the limitation part.

**Clarity:**

Paper is well written. Adequate citation. Motivation is clear and so are the experiments.

**Correctness:**

The definition of OOD is not in a sound way, which should be addressed by the authors.

**Documentation:**

Well documented.

**Limitations:**

No limitations section in the paper. The major concern for me is the method used for collecting OOD data. Normally, we prepare OOD data by some designed “natural” definitions, e.g., we collect images from different hospitals. In this way, the collected OOD data is meaningful. However, this paper uses pairs similarity and model degradation to guide the OOD collection which is highly model-dependent. Different models used to conduct similarity analysis (in this paper, SimCSE is used) will produce different scores for the same pairs, in turn, results in different ID/OOD data splitting. The same problem could happen in model degradation analysis. Therefore, I am not sure of the usefulness of the prepared datasets in this work.

**Opportunities For Improvement:**

- Rigorous: some takeaways need to exclude some exception cases and there is no related explanation of exception values. i.e. Evaluation 4.2 takeway3, some ICL of OOD show worse performance than ICL of ID (SST of davinci3 shows 73.95 against 77.23, same for LLaMA), some ICL of OOD even show worse than 0-shot (LLaMA DS gets 52.26 against 54.05).
- Lack of analysis of different distribution shifts: this work provides three types of ID-OOD correlation, but it shows no discussion about the detailed OOD distribution shift types (adversarial/debiase/…)

**Relation To Prior Work:**

The difference between this work and previous work is clearly discussed.

**Summary And Contributions:**

This work focuses on the robustness of OOD in NLP. It evaluates various models and methods for five different tasks. Besides, it proposes a novel ID-OOD data analysis method and summarizes some important patterns. Overall, this work holds significant importance in the research of OOD problems in NLP. It provides new insights, novel ideas, and a good reusable benchmark that can guide future research endeavors.

---

> ### Author Response · Authors · 2023-08-13
> **Response to Reviewer 6XNi**
>
> # Opportunities For Improvement
> 1. Some takeaways need to exclude some exception cases and there is no related explanation of exception values.
>
> **A:** Sorry for missing the explanation. According to [1], in-context demonstrations aim to guide the models to learn the target label space, instead of the feature-label mapping. The ID examples contain more diverse information due to the construction process of our benchmark. Thus, ID examples can better prompt the language models to locate the target label space, compared to the OOD examples that may target a specific domain.
>
> [1] Rethinking the Role of Demonstrations: What Makes In-Context Learning Work? Min et al. EMNLP 2022.
>
> 2. Lack of analysis of different distribution shifts
>
> **A:** In benchmark construction, we did not select distribution shifts by "types". We included adversarial shifts except for NER in which there are no publicly available adversarial datasets, but results show that adversarial shifts do not differ significantly from other shifts. However, for other types, there may not consistently exist datasets of the same type across all tasks. For example, only SA and NLI have debiased datasets (IMDb-cont / HANS), while the other tasks do not. Hence, it is hard to analyze a type of distribution shift systematically.
>
> # Limitations
> 1. This paper uses pairs similarity and model degradation to guide the OOD collection which is highly model-dependent.
>
> **A:** To address your concern for SimCSE, we try other alternatives in this discussion period. The results are as follows:
>
> |SimCSE||||||||
> |---|---|---|---|---|---|---|---|
> |source \|target|Amazon|DSC|Dynasent|IMDb|SemEval|SST|Yelp|
> |Amazon|100.00|86.02|57.30|36.67|24.74|33.70|49.22|
> |DSC|86.02|100.00|59.15|54.55|31.70|44.40|55.45|
> |Dynasent|57.30|59.15|100.00|32.69|28.17|19.68|88.99|
> |IMDb|36.67|54.55|32.69|100.00|46.95|84.62|39.88|
> |SemEval|24.74|31.70|28.17|46.95|100.00|40.45|24.03|
> |SST|33.70|44.40|19.68|84.62|40.45|100.00|19.43|
> |Yelp|49.22|55.45|88.99|39.88|24.03|19.43|100.00|
>
> |ESimCSE||||||||
> |---|---|---|---|---|---|---|---|
> |source \|target|Amazon|DSC|Dynasent|IMDb|SemEval|SST|Yelp|
> |Amazon|100.00|95.27|88.46|76.07|76.50|76.19|81.75|
> |DSC|95.27|100.00|84.22|80.06|76.68|80.97|79.26|
> |Dynasent|88.46|84.22|100.00|70.64|79.28|74.41|90.52|
> |IMDb|76.07|80.06|70.64|100.00|72.16|83.73|72.82|
> |SemEval|76.50|76.68|79.28|72.16|100.00|76.75|68.64|
> |SST|76.19|80.97|74.41|83.73|76.75|100.00|62.44|
> |Yelp|81.75|79.26|90.52|72.82|68.64|62.44|100.00|
>
> |Sentence-Transformer||||||||
> |---|---|---|---|---|---|---|---|
> |source \|target|Amazon|DSC|Dynasent|IMDb|SemEval|SST|Yelp|
> |Amazon|100.00|90.25|46.57|48.62|30.65|48.92|49.23|
> |DSC|90.25|100.00|37.27|61.54|25.28|54.79|49.49|
> |Dynasent|46.57|37.27|100.00|24.76|36.94|32.65|78.37|
> |IMDb|48.62|61.54|24.76|100.00|25.22|84.26|34.05|
> |SemEval|30.65|25.28|36.94|25.22|100.00|40.66|19.26|
> |SST|48.92|54.79|32.65|84.26|40.66|100.00|24.58|
> |Yelp|49.23|49.49|78.37|34.05|19.26|24.58|100.00|
>
>
>
> From the results, we can see that despite showing some variance, the trends are consistent. Certain datasets are similar to ID datasets whatever the semantic representation model used, and vice versa. Since we only care about the relative ranking of the similarity between each OOD dataset to the ID dataset, using different models will give the same result.
>
> As for performance degradation, according to the results in Section 3, the ID-OOD performance of different models on the same dataset is strongly correlated, so the trend of performance degradation of different models, which have similar ID performance on each OOD, should be similar. It is predictable that different models would lead to similar orders of performance degradation on OOD datasets,, so we can use different models to get consistent results.
>
> # Correctness
> 1. The definition of OOD is not in a sound way.
>
> **A:** Sorry for missing the definition. In our work, we can define distribution shifts considered in our benchmark from two perspectives.
> Firstly, [1] classifies distribution shifts into "semantic shift" and "background shift." Our use of "out-of-distribution" aligns with the concept of "background shift," which involves changes in the domain or style of the text while preserving the semantic content.
> Secondly, [2] formally defines three types of distribution shifts: covariate shift, label shift, and concept shift. In our work, we mainly focus on the combination of covariate shifts and concept shifts. This indicates that the model needs to generalize well to different input features (a.k.a, covariate shift) and adapt to variations in the underlying concepts within the data (a.k.a, concept shift).
>
> [1] Types of Out-of-Distribution Texts and How to Detect Them. Arora et al. EMNLP 2021.
>
> [2] Dive into Deep Learning. Zhang et al.

---

> ### Author Response · Authors · 2023-08-26
> **Hope to receive your reply**
>
> Dear reviewer,
>
> Thanks for your efforts and your insightful suggestions on this paper. We have tried our best to address your concerns and have incorporated your advice to improve the paper. If possible, could you please kindly take a look at our response? Please also let us know if there is any further question about our work, thanks!
>
> Best regards,
>
> Authors

---

> ### Comment · Area_Chair_yQb7 · 2023-08-29
>
> Dear Reviewer
>
> Kindly review and reply to the feedback provided by the authors.
>
> Regards
> AC

---

### Official Review · Reviewer_wTiN · 2023-07-20
**BOSS benchmark**

**Rating:** 6
**Confidence:** 3

**Strengths:**

- The analysis of ID-OOD pair similarity is useful for understanding model performance on OOD data. However, the paper claims that in prior work ID-OOD pairs sometimes come from the same source (or distribution), but only one example of that is given.
- The paper attempts to establish criteria for what makes a valuable ID-OOD pair. This is an important step in pushing the field forward towards being less arbitrary in the creation/choice of evaluation benchmarks.
- Inclusion of LLMs in analysis. The paper goes beyond straightforward "vanilla" evaluation.

**Additional Feedback:**

- Some references use the arxiv version when the newer published version may be preferred. For instance, [1] uses the arxiv version, but there is a newer published version: https://aclanthology.org/2022.tacl-1.53/.

**Clarity:**

- "Despite OOD robustness in NLP has been extensively studied" (Line 218) seems to be missing a word.
- Information essential to understanding Section 3 and Figure 1 is relegated to the appendix. I found Figure 1 to be difficult to understand.

**Correctness:**

(I lump general questions and 'correctness' concerns here)

- C1) The paper says that prior work choses datasets based on "popularity", but it appears that this paper does the same thing when selecting sentiment analysis benchmarks in Section 2.3. I find it hard to believe that these seven listed are the only sentiment analysis datasets on Kaggle/paperswithcode/acl-anthology since 2010. Or were these the only ones that were filtered by the diversity protocol from Section 2.2?
- C2) Similar to the above, "driven by popularity" (e.g., line 372) is not qualified/quantified in any way by the paper. What exactly does this mean? Many of the datasets included in BOSS are popular datasets. Others are often either (1) brand new (AdvCivil) or (2) published within the past 1.5 years. On the other hand, there may be many 'un-popular' (my term, not yours) datasets that are not included in BOSS but could be. Similarly line 266 refers to some evaluations as "non-standard", which is not defined.
- C3) I like that the paper has discussion of label mapping (Section B.2.4). However, for NER at least, I wonder if some datasets annotate certain spans differently than others. For instance, "The New York Times" could have "New York Times" labeled as an organization in Dataset A, but "The" could be included in the organization annotation span in Dataset B. Alternatively (and in my opinion erroneously) "New York" could be a location in Dataset C. Such cases could lead to low performance if a model is trained on Dataset A but tested on Datasets B and C.
- C4) Can you explain how the citation #7 (line  44) supports the third criteria for selecting ID and OOD data? (Same for line 108.)
- C5) Why was SimCSE chosen over alternatives?
- C6) Dataset pairs with high similarity were removed from consideration, but why not just apply adversarial filtering to the samples of each ID-OOD dataset pair?
- C7) "The ID dataset should provide sufficient knowledge for models to handle the task" (line 95). The paper uses dataset size as a proxy for "sufficient knowledge", but that is a stretch. There is no guarantee that dataset size provides more knowledge. Similarly, "diversity" (line 99) is not quantified.
- C8) BOSS is created by filtering out dataset pairs that do not exhibit enough performance degradation. As a practitioner/engineer, I would want access to something like Table 3, which provides results for 7 scores, as opposed to just 3 scores. This might be my biggest concern. Why not just include all ID-OOD pairs, and let the practitioner decide which are important?

**Documentation:**

n/a

**Limitations:**

- Some NLP researchers/practitioners may feel that the tasks included in BOSS are not representative of all of NLP. Indeed, one could accuse the inclusion of tasks in BOSS as being driven by popularity.

**Opportunities For Improvement:**

- The main concept/term of the paper, "out-of-distribution", is never defined in the paper. I think it is worth a couple sentences explaining what exactly is meant by "out-of-distribution". For this paper, it is in-domain distribution-shifted data. For other applications, "out-of-distribution" might mean "out-of-scope" (e.g., https://aclanthology.org/D19-1131.pdf) or "out-of-domain" (e.g., an MNIST digit classifier encountering an image of the letter "t"). Perhaps using "background" and "semantic" shift terminology from https://aclanthology.org/2021.emnlp-main.835.pdf would be helpful to the reader.
- I find the claim of "This paper reexamines the research on out-of-distribution (OOD) robustness in the field of NLP. We find that the distribution shift settings in previous studies commonly lack adequate challenges, hindering the accurate evaluation of OOD robustness" to be a weak one. Evidence for this claim is relegated to a table (Table 7, with an extremely vague caption of "Survey" and no indication of what  the arrows mean) and a three sentence paragraph (Section A). The paper continues this argument with "We observe a lack of standardized OOD benchmark suites tailored for NLP resulting in the adoption of heuristic and popularity-based dataset selection strategies in previous work." However, it is not clear what is meant by "standardized OOD benchmark suites tailored for NLP" or "popularity-based dataset selection strategies" (see C2 below).

**Relation To Prior Work:**

- Line 23 "existing evaluation assumes independent and identically distributed (i.i.d) condition": I think this should include the word "often". There is a growing body of work that criticizes this i.i.d. condition. For instance, https://aclanthology.org/2021.eacl-main.156v2.pdf, and the 2nd paragraph of Section 2.2 of this paper: https://aclanthology.org/2023.eacl-main.195.pdf.

**Summary And Contributions:**

This paper introduces BOSS, a suite of ID-OOD dataset pairs for 5 NLP tasks. The paper proposes several criteria for selecting useful ID-OOD dataset pairs: 1) the ID dataset should be large/diverse enough for a model to learn sufficient knowledge, 2) ID and OOD pairs should be sufficiently different, and 3) OOD shifts should be challenging. The paper proposes to filter candidate ID-OOD pairs according to this criteria by using semantic similarity based methods for the 2nd criteria and performance scores (on a model trained on ID and tested on OOD) for the 3rd criteria. This is in contrast to prior work, which appears to select ID-OOD pairs by convenience; in some cases, the ID-OOD pairs in prior work are actually derived from similar sources, making the OOD performance potentially inflated. The paper then benchmarks pre-trained language models and LLMs on BOSS using several off-the-shelf methods for attempting to improve OOD performance.

---

> ### Author Response · Authors · 2023-08-13
> **Response to Reviewer wTiN (1/4)**
>
> # Strengths
> 1. The analysis of ID-OOD pair similarity is useful for understanding model performance on OOD data. However, the paper claims that in prior work ID-OOD pairs sometimes come from the same source (or distribution), but only one example of that is given.
>
> **A:** Thanks for your recognition.
>
> We did not explicitly showcase more references, but in fact, many other examples can be found in Table 7.
> For sentiment analysis, many papers (citation #44, #38, #69, #98, #14, #92, #71, #100) adopt SST-IMDb or different subsets of Amazon as ID-OOD pairs.
> For extractive question answering, a lot of works (citation #29, #81, #104, #50, #102) consider using SQuAD, HotpotQA, and Trivia-QA, which are all collected from Wikipedia, to perform OOD evaluations.
>
> # Opportunities For Improvement
>
> 1. The main concept/term of the paper, "out-of-distribution", is never defined in the paper.
>
> **A:** Sorry for missing further explanations. There exist multiple definitions of OOD in literature[1,2], and we can define distribution shifts considered in our benchmark from two perspectives.
> Firstly, [1] classifies distribution shifts into "semantic shift" and "background shift." Our use of "out-of-distribution" aligns with the concept of "background shift," which involves changes in the domain or style of the text while preserving the semantic content.
> Secondly, [2] formally defines three types of distribution shifts: covariate shift, label shift, and concept shift. In our work, we mainly focus on the combination of covariate shift and concept shift. This indicates that the model needs to generalize well to different input features (a.k.a, covariate shift) and adapt to variations in the underlying concepts within the data (a.k.a, concept shift).
>
> [1] Types of Out-of-Distribution Texts and How to Detect Them. Arora et al. EMNLP 2021.
>
> [2] Dive into Deep Learning. Zhang et al.
>
> 2. I find the claim of "We find that the distribution shift settings in previous studies commonly lack adequate challenges, hindering the accurate evaluation of OOD robustness" to be a weak one. Evidence for this claim is relegated to a table (Table 7, with an extremely vague caption of "Survey" and no indication of what the arrows mean) and a three sentence paragraph (Section A). The paper continues this argument with "We observe a lack of standardized OOD benchmark suites tailored for NLP resulting in the adoption of heuristic and popularity-based dataset selection strategies in previous work." However, it is not clear what is meant by "standardized OOD benchmark suites tailored for NLP" or "popularity-based dataset selection strategies" (see C2 below).
>
> **A:** Apologies for any misunderstandings. Our survey is grounded in two recognized taxonomy papers. We meticulously reviewed **all** references and documented papers addressing OOD robustness across the five tasks in question. The first column in Table 7 of our paper signifies individual papers, the second column details the evaluated tasks, and the third column lists the encompassed datasets. Our intention was to represent dataset settings as ID dataset $\rightarrow$ OOD datasets. We regret that the format oversight led to inconsistencies, causing reader confusion. We will rectify this in the revision.
>
> Through the survey, we can observe that there is no single uniform set of datasets for evaluation in NLP, namely no “standardized OOD benchmark suites tailored for NLP”. And we can find that existing work includes evaluation datasets without particular consideration while mainly directly using popular datasets, i.e. adopting "popularity-based dataset selection strategies".

---

> ### Author Response · Authors · 2023-08-13
> **Response to Reviewer wTiN (2/4)**
>
> # Limitations
> 1. The tasks included in BOSS are not representative of all of NLP. Indeed, one could accuse the inclusion of tasks in BOSS as being driven by popularity.
>
> **A:** We agree that these five tasks may not encompass the entirety of NLP, given the extensive range of tasks examining various aspects of model performance. The chosen tasks within BOSS evaluate models across natural language understanding, structured data prediction, and question answering, which are core language model competencies.
>
> Also, in constructing our benchmark, we considered extending other tasks such as NLG and commonsense reasoning, but in practice, we found a lot of difficulties.
>
> For NLG tasks, the primary concern lies in evaluation. Since different datasets exhibit distinct text styles, an in-domain (ID) model might generate responses with styles diverging from the reference answers when tested on out-of-domain (OOD) datasets. However, current NLG metrics evaluate predictions based on their resemblance to reference answers, which might not accurately reflect text quality in scenarios with a vast output space [3]. An emerging alternative involves employing Language Models like GPT-4 for scoring predictions. However, this method is cost-intensive and lacks reproducibility due to unpredictable updates. Considering these evaluation challenges, including generation tasks within this OOD benchmark might not be appropriate.
>
> For commonsense reasoning, multiple datasets exist from various sources, each demanding distinct knowledge. These differences are substantial enough that knowledge gained from an ID dataset might not be applicable to OOD datasets. For instance, HellaSwag [4] necessitates basic world knowledge and logical reasoning to complete a sentence, while StepGame [5] relies on spatial imagination without requiring world knowledge. The dissimilarity in required abilities suggests that models should acquire knowledge through pre-training rather than fine-tuning on an ID dataset and then transferring it to OOD tasks. This approach aligns more reasonably with the task's nature.
>
> [3] News Summarization and Evaluation in the Era of GPT-3. Goyal et al. 2022.
>
> [4] HellaSwag: Can a Machine Really Finish Your Sentence? Zellers et al. ACL 2019.
>
> [5] StepGame: A New Benchmark for Robust Multi-Hop Spatial Reasoning in Texts. Shi et al. AAAI 2022.
>
> # Correctness
> 1. Were these seven sentiment analysis datasets listed only ones that were filtered by the diversity protocol from Section 2.2?
>
> **A:** We did pre-filter some datasets for specific reasons. Taking sentiment analysis as an example, among the 87 datasets available on `paperswithcode`, many are non-English or not focused on sentence classification. Additionally, while `Kaggle` offers numerous datasets, only 36 of them are non-commercial, and most lack quality guarantees as they are released by unknown individuals.
>
> After this filtering, our inclusion prioritizes ternary classification datasets due to their heightened complexity compared to binary counterparts. Notably, this filters out a majority of sentiment analysis datasets. We retain IMDb in our experiments to demonstrate limitations in prior benchmarks. Certain datasets like Amazon Fine Foods [6] are omitted because of substantial overlap with our Amazon dataset but are less diverse (our Amazon dataset covers 29 product categories, while Amazon Fine Foods centers solely on food products). Furthermore, datasets like RETWEET [7] provide only text IDs rather than full text due to licensing constraints. Such datasets don't meet our criterion of public availability, as they compromise user convenience.
>
> In short, a lot of prior efforts have been made. Due to the limited space, we are not able to explain every detail in the paper, but we did not adopt the criteria of "popularity" in the selection of datasets.
>
> [6] From Amateurs to Connoisseurs: Modeling the Evolution of User Expertise through Online Reviews. McAuley et al. WWW 2013.
>
> [7] How Will Your Tweet Be Received? Predicting the Sentiment Polarity of Tweet Replies. Arasteh et al. ICSC 2015.

---

> ### Author Response · Authors · 2023-08-13
> **Response to Reviewer wTiN (3/4)**
>
> 2. "driven by popularity" (e.g., line 372) is not qualified/quantified in any way by the paper. Similarly line 266 refers to some evaluations as "non-standard", which is not defined.
>
> **A:** By "driven by popularity," we do not imply a wholesale exclusion of popular datasets or a deliberate avoidance of them. Rather, we refer to instances where researchers incorporate datasets into OOD benchmarks without comprehensive explanation, seemingly opting for a somewhat arbitrary selection. Our survey reveals that this practice often results in the adoption of popular datasets. In this paper, our inclusion of "popular datasets" is based on their alignment with our criteria, not their popularity. These two selection strategies stem from distinct motivations and can potentially yield different evaluation outcomes.
>
> When we use the term "non-standard," we mean that dataset selections lack uniformity and do not adhere to a specific protocol, often relying on the direct inclusion of popular datasets during evaluation.
>
> 3. For NER at least, I wonder if some datasets annotate certain spans differently than others.
> **A:** Thanks for raising this issue. In regard to dataset C, the situation you highlighted could indeed be attributed to an annotation error, wherein "New York" should not have been separated from "Times" and labeled as a "location." This aspect pertains to the dataset's inherent quality. Unfortunately, we lack a feasible solution for such cases, as manually scrutinizing each sample appears impractical within our scope. Concerning the misalignment of tags for the "The" token, this can also be considered as inevitable noise.
>
> 4. Can you explain how the citation #7 (line 44) supports the third criteria for selecting ID and OOD data?
>
> **A:** In the third paragraph of section 3.3 in citation #7, the authors claimed that current models evolve rapidly, hence more difficult datasets are required for a reliable evaluation ("Since our systems continue to improve rapidly, though, we should expect to be spending more time in the long tail of our data difficulty distributions: If we build reliable datasets, much of their future value may lie in their ability to measure improvements in accuracy among highly accurate systems."  ). Also, in Figure 1, the authors noted that "Benchmark datasets need to be much harder and/or much larger." Therefore, in line 3, we claim that existing benchmark datasets deviate from this expectation for benchmark difficulty, thus motivating us to design the third criterion.
>
> 5. Why was SimCSE chosen over alternatives?
>
> **A:** To the best of our knowledge, SimCSE and sentence-BERT are the most widely used models and SimCSE is more advanced according to the SimCSE paper, hence we choose SimCSE as the semantic representation model. In this discussion period, we also try the other variants of SimCSE, ESimCSE [8], and the best model in sentence-transformers leaderboard (`all-mpnet-base-v2`). The results are as follows:
>
> |SimCSE||||||||
> |---|---|---|---|---|---|---|---|
> |source \|target|Amazon|DSC|Dynasent|IMDb|SemEval|SST|Yelp|
> |Amazon|100.00|86.02|57.30|36.67|24.74|33.70|49.22|
> |DSC|86.02|100.00|59.15|54.55|31.70|44.40|55.45|
> |Dynasent|57.30|59.15|100.00|32.69|28.17|19.68|88.99|
> |IMDb|36.67|54.55|32.69|100.00|46.95|84.62|39.88|
> |SemEval|24.74|31.70|28.17|46.95|100.00|40.45|24.03|
> |SST|33.70|44.40|19.68|84.62|40.45|100.00|19.43|
> |Yelp|49.22|55.45|88.99|39.88|24.03|19.43|100.00|
> |ESimCSE||||||||
> |source \|target|Amazon|DSC|Dynasent|IMDb|SemEval|SST|Yelp|
> |Amazon|100.00|95.27|88.46|76.07|76.50|76.19|81.75|
> |DSC|95.27|100.00|84.22|80.06|76.68|80.97|79.26|
> |Dynasent|88.46|84.22|100.00|70.64|79.28|74.41|90.52|
> |IMDb|76.07|80.06|70.64|100.00|72.16|83.73|72.82|
> |SemEval|76.50|76.68|79.28|72.16|100.00|76.75|68.64|
> |SST|76.19|80.97|74.41|83.73|76.75|100.00|62.44|
> |Yelp|81.75|79.26|90.52|72.82|68.64|62.44|100.00|
> |Sentence-Transformer||||||||
> |source \|target|Amazon|DSC|Dynasent|IMDb|SemEval|SST|Yelp|
> |Amazon|100.00|90.25|46.57|48.62|30.65|48.92|49.23|
> |DSC|90.25|100.00|37.27|61.54|25.28|54.79|49.49|
> |Dynasent|46.57|37.27|100.00|24.76|36.94|32.65|78.37|
> |IMDb|48.62|61.54|24.76|100.00|25.22|84.26|34.05|
> |SemEval|30.65|25.28|36.94|25.22|100.00|40.66|19.26|
> |SST|48.92|54.79|32.65|84.26|40.66|100.00|24.58|
> |Yelp|49.23|49.49|78.37|34.05|19.26|24.58|100.00|
>
> From the results, despite different methods varying to some extent, the trend of similarity almost remains unchanged. Therefore, we consider that the essential is to adopt an advanced semantic representation model, and that which particular model we choose will not lead to differences in our selection.
>
> [8] ESimCSE: Enhanced Sample Building Method for Contrastive Learning of Unsupervised Sentence Embedding. Wu et al. COLING 2022.

---

> ### Author Response · Authors · 2023-08-13
> **Response to Reviewer wTiN (4/4)**
>
> 6. Dataset pairs with high similarity were removed from consideration, but why not just apply adversarial filtering to the samples of each ID-OOD dataset pair?
>
> **A:** Applying adversarial filtering to ensure the distinction of samples requires measuring the distance between a sample and a distribution, and then determining whether a sample should be retained. However, it is difficult to set an absolute value threshold for whether to retain samples. Instead, in our current pipeline, we make decisions by the ranking of similarity. We only need to filter out datasets that are the most similar to the ID dataset and leave about four or five datasets for the next step.
>
> 7. "The ID dataset should provide sufficient knowledge for models to handle the task" (line 95). The paper uses dataset size as a proxy for "sufficient knowledge", but that is a stretch. There is no guarantee that dataset size provides more knowledge. Similarly, "diversity" (line 99) is not quantified.
>
> **A:** This criterion for ID datasets is consistent with [9] (abstraction) and [10] (section 1, page 4), which found that training models on large and diverse datasets improve model robustness to distribution shifts.
> We believe that large datasets do not naturally contain sufficient knowledge, but datasets that are both large and diverse may do with a high probability. A diverse dataset consists of various patterns of the task, and the large data size allows the dataset to statistically contain a sufficient number of samples with different patterns for the model to learn from.
> For "diversity", it is hard to be quantified by conducting experiments. Instead, we consider the collection process of each dataset, including the number of sources and subtypes (e.g., styles, topics, levels of formality, etc.).
>
> [9] Measuring Robustness to Natural Distribution Shifts in Image Classification. Taori et al. NeurIPS 2020.
> [10] Accuracy on the Line: On the Strong Correlation Between Out-of-Distribution and In-Distribution Generalization. Miller et al. ICML 2021.
>
>
> 8. Why not just include all ID-OOD pairs, and let the practitioner decide which are important?
>
> **A:** Our primary motivation for undertaking this endeavor is to cater not only to experts well-versed in this field, who possess the necessary expertise to critically evaluate the reasonability of results, but also to practitioners who may be less familiar with OOD problems or this particular benchmark. Our aim is to offer a benchmark that assists practitioners of all level in developing their models effectively.
>
> However, we contend that not all the scores provided are equally useful, and some results may even be misleading for part of the target audience. For instance, if we consider Table 3 and focus solely on the three filtered scores, practitioners will readily discern that the OOD problem in sentiment analysis is still not adequately addressed. However, when all seven scores in Table 3 are retained, it becomes apparent that the ID model achieves an accuracy of nearly 90% on 50% of the distribution shifts. Despite this, we emphasize that the scores that are meaningful for evaluation remain limited to the three initially filtered ones. The presence of additional distractors in the results could potentially mask the overall performance, thereby leaving practitioners unaware of the severity of OOD problems.
>
> As we scale up the dataset pool and incorporate four other datasets where the ID model performs well, the significance of the scores that are meaningful for evaluation appears to be further diluted and less discernible. Consequently, there is a risk that some practitioners might overlook these three datasets as special outliers. Moreover, if individuals independently collect candidate datasets without including the ones that are meaningful for evaluation in the pool, the evaluation could lead to a wholly misleading assessment.
> Hence, it becomes imperative to furnish practitioners with a comprehensive and rigorous set of datasets in advance, ensuring that the benchmark remains meaningful and insightful for their model development efforts.
>
> # Clarity
> 1. Information essential to understanding Section 3 and Figure 1 is relegated to the appendix. I found Figure 1 to be difficult to understand.
>
> **A:** Sorry for the confusion. We would address the issue in our revision.
>
> # Relation To Prior Work
> 1. Line 23 "existing evaluation assumes independent and identically distributed (i.i.d) condition": I think this should include the word "often".
>
> **A:** Thanks for the suggestion! We would make the wording more precise in the next version.
>
> # Additional Feedback
> 1. Some references use the arxiv version when the newer published version may be preferred.
>
> **A:** Thanks for your careful reading! We will double-check all references carefully.

---

> ### Author Response · Authors · 2023-08-23
> **Hope to receive your reply**
>
> Dear reviewer,
>
> Thanks for your efforts and your insightful suggestions on this paper. We have tried our best to address your concerns and have incorporated your advice to improve the paper. If possible, could you please kindly take a look at our response? Please also let us know if there is any further question about our work, thanks!
>
> Best regards,
>
> Authors

---

> ### Comment · Area_Chair_yQb7 · 2023-08-29
>
> Dear Reviewer
>
> Kindly review and reply to the feedback provided by the authors.
>
> Regards
> AC

---

### Official Review · Reviewer_XEXa · 2023-07-21
**Novel framework for IID and OOD data selection but still old datasets - Boderline accept**

**Rating:** 6
**Confidence:** 5
**Clarity:** The paper is easy to follow however t…

**Strengths:**

**Novel Framework**

The paper proposes a novel framework for selecting IID and OOD datasets. This framework can be used to evaluate the generalization capabilities of LLMs, compare various small finetuned language model architecture on OOD data and benchmark the robustness methods that works well.

**OOD Experiments**

The paper has extensive experiments that compares how IID data, tunabale parameters and training steps affects OOD performance and benchmarks recent NLP robustness methods on the proposed BOSS framework.

The paper has experiments comparing zeroshot, IID and OOD in-context learning in the case of LLMs and discuss tradeoffs when small fine-tuned language models perform better compared to LLMs.

**Additional Feedback:**

I have added my feedback in the Opportunities For Improvement section of the review.

**Correctness:**

The claims made in the paper are correct, the dataset has been constructed in a sound manner, the evaluation methods and experiment designs are appropriate and performed correctly.



**Documentation:**

There are sufficient detail on data collection and organization, availability and maintenance, and ethical and responsible use.

There is sufficient detail to support reproducibility.

**Ethics:**

No, there are no ethical concerns.

**Limitations:**

The authors adequately addressed the limitations and potential negative societal impact of their work.

**Opportunities For Improvement:**

**OOD Dataset Selection**

The paper proposed methods for IID and OOD dataset selection for each task. Although the assumptions used for IID dataset selection seems relevant, some of the assumptions used for OOD dataset are not relevant. The authors look at the performance drop of IID to OOD on T5, dataset level semantic similarity and the source or distribution of a dataset to select the final OOD datasets. Although, considering the source of dataset for OOD selection is fine, but there is no correlation between dataset semantic similarity and IID to OOD performance drop. As an example similarity between Amazon and DynaSent is ~57.0 but the OOD performance is low ~48, and similarity between Amazon and IMDB is even lower ~36 but the OOD performance is high ~92. Similar observations can be made for NLI and Question answering tasks. Also, the method used to calculate dataset level similarity is based upon taking the average of the RoBERTa embeddings of all the tokens in a dataset and computing the dot product with the embedding vector of other dataset. This also seems not the right method as simply averaging all the tokens will introduce lot of noise and datasets with more tokens might have a very different centroid embedding vector.  Is there any prior work that computes dataset level similarity in such a manner? If yes, that should be included in the related work.

**Data Leakage**

Although the proposed IID and OOD dataset pairs are fine for comparing small fine-tuned language models, it makes no sense to compare LLMs on public benchmarks, as the test sets of these datasets might be part of the pre-training datasets of LLMs (Line 341 - 345). Because of this the comparison in Table 6 between (T0, T5) and (LLaMa, Davinci, Turbo) is uselesss as there is no way to verify data leakage. The results only make sense if one can verify that the OOD datasets are not part of the pre-training corpus of LLMs. Therefore, I believe the proposed framework of IID and OOD selection is fine but one needs new benchmark datasets for each task.

**NLP tasks**

Certain NLP tasks like sentiment analysis are trivial given the generalization capabilities of LLMs. This calls for more novel and harder tasks to evaluate LLMs. There has been a lot of progress in developing novel models in the past 7 or 8 years (RNNs to GPT), however the NLP commulity is still using the 8-10 year old benchmarks to evalauate LLMs. As the models are becoming stronger, we need even stronger NLP benchmarks and tasks.


**Relation To Prior Work:**

The authors have clearly discussed prior work and how the proposed benchmark is a novel contribution.

**Summary And Contributions:**

The paper proposes BOSS a new benchmark for Out of Distribution (OOD) evaluation of pretrained language models. The benchmark consists of 5 different tasks: Sentiment analysis, Natural Language Inference, Toxic content detection, Named Entity Recognition and Question Answering. Each task has a large in-distribution dataset (IID) that is used for fine-tuning pretrained models and several OOD datasets.
The paper has extensive experiments that compare small fine-tuned langauge models across various dimensions and discuss tradeoffs between small and large language models (LLMs) such as LLaMa, GPT-3.5.

Overall, the paper proposes a nice framework and have rigorous experiments and analysis, however I believe the datasets used are quite old and might already be part of the pre-training datasets of LLMs.

---

> ### Author Response · Authors · 2023-08-13
> **Response to Reviewer XEXa (1/2)**
>
> # Opportunities For Improvement
> 1. Although, considering the source of dataset for OOD selection is fine, but there is no correlation between dataset semantic similarity and IID to OOD performance drop.
>
> **A:** Apologies for any lack of clarity. Our selection protocol is designed to address two aspects of distribution shifts: distinction and difficulty. We employ corresponding metrics for measurement, namely utilizing SimCSE to assess distinction (Line 104-105) and gauging performance drop for difficulty (Line 109-110). It's important to note that these two metrics do not necessarily exhibit correlation, as they serve distinct objectives.
>
> 2. the method used to calculate dataset level similarity is based upon taking the average of the RoBERTa embeddings of all the tokens in a dataset and computing the dot product with the embedding vector of other dataset. This also seems not the right method as simply averaging all the tokens will introduce lot of noise and datasets with more tokens might have a very different centroid embedding vector. Is there any prior work that computes dataset level similarity in such a manner?
>
> **A:**Thanks for pointing that out. We refer to classical distance calculation methods used in clustering: https://en.wikipedia.org/wiki/Hierarchical_clustering. There are seven methods in total, “single” (minimal distance), “complete” (maximal distance), “average”, “weighted”, “centroid”, “median”, and “ward”.
>
> Notably, "weighted" and "ward" necessitate a third cluster, and "single" and "complete" methods rely on two data points to signify cluster distance, so we exclude the four methods. Moreover, the "median" method bears a resemblance to the "centroid". Consequently, we retain two methods: "centroid" and "average." We opt for "centroid" due to its potential to yield a distance of 0 between a dataset and itself, which looks better for results presentation. We do, however, supplement our study with an experiment involving the use of the "average" method to compute dataset-level similarity through pairwise distance averaging. The results are as follows:
>
> | centroid |  |  |  |  |  |  |  |
> |---|---|---|---|---|---|---|---|
> | source \ target | Amazon | DSC | Dynasent | IMDb | SemEval | SST | Yelp |
> | Amazon | 100.00 | 86.02 | 57.30 | 36.67 | 24.74 | 33.70 | 49.22 |
> | DSC | 86.02 | 100.00 | 59.15 | 54.55 | 31.70 | 44.40 | 55.45 |
> | Dynasent | 57.30 | 59.15 | 100.00 | 32.69 | 28.17 | 19.68 | 88.99 |
> | IMDb | 36.67 | 54.55 | 32.69 | 100.00 | 46.95 | 84.62 | 39.88 |
> | SemEval | 24.74 | 31.70 | 28.17 | 46.95 | 100.00 | 40.45 | 24.03 |
> | SST | 33.70 | 44.40 | 19.68 | 84.62 | 40.45 | 100.00 | 19.43 |
> | Yelp | 49.22 | 55.45 | 88.99 | 39.88 | 24.03 | 19.43 | 100.00 |
>
> | average |  |  |  |  |  |  |  |
> |---|---|---|---|---|---|---|---|
> | source \ target | Amazon | DSC | Dynasent | IMDb | SemEval | SST | Yelp |
> | Amazon | 19.68 | 17.75 | 8.35 | 10.57 | 3.66 | 7.14 | 12.27 |
> | DSC | 17.75 | 21.65 | 9.02 | 16.48 | 4.91 | 9.82 | 14.49 |
> | Dynasent | 8.35 | 9.02 | 10.74 | 6.96 | 3.08 | 3.11 | 16.39 |
> | IMDb | 10.57 | 16.48 | 6.96 | 42.10 | 10.12 | 26.00 | 14.53 |
> | SemEval | 3.66 | 4.91 | 3.08 | 10.12 | 10.96 | 6.37 | 4.49 |
> | SST | 7.14 | 9.82 | 3.11 | 26.00 | 6.37 | 22.46 | 5.19 |
> | Yelp | 12.27 | 14.49 | 16.39 | 14.53 | 4.49 | 5.19 | 31.57 |
>
> Despite numerical disparities, both methods exhibit a congruent similarity trend. Hence, we assert that the universality of the results remains unaffected.
>
> 3. Data Leakage
>
> **A:** We totally agree with the standpoint of calling for new benchmarks, as we stated in line 346-347. Currently we do not have a good solution to address the data leakage issue due to the lack of brand new datasets in NLP. However, with our established protocols, we can easily evolve the benchmark as new datasets come out.
>
> In addition, we may hold respectfully different view with the claim that "the comparison in Table 6 between (T0, T5) and (LLaMa, Davinci, Turbo) is uselesss". Since LLMs are possible to cheat in the comparison, it is true that it remains unsure whether the advantage on OOD of LLMs is attributed to their superior generalizability or to their memorization of the answers. However, LLMs still fall behind small models on ID datasets, which further reinforces our first takeaway that "Fine-tuning small domain-specific models is superior when enough training data is available".

---

> ### Author Response · Authors · 2023-08-13
> **Response to Reviewer XEXa (2/2)**
>
> # Opportunities For Improvement
> 4. Certain NLP tasks like sentiment analysis are trivial given the generalization capabilities of LLMs. This calls for more novel and harder tasks to evaluate LLMs.
>
> **A:** According to the results in Table 6, it seems that sentiment analysis is relatively less challenging compared to other tasks. However, there is still much room for improvement in the performance of LLMs.
>
> Also, in constructing our benchmark, we considered extending harder tasks such as NLG and commonsense reasoning, but in practice, we found a lot of difficulties.
>
> - For NLG tasks, the primary concern lies in evaluation. Since different datasets exhibit distinct text styles, an in-domain (ID) model might generate responses with styles diverging from the reference answers when tested on out-of-domain (OOD) datasets. However, current NLG metrics evaluate predictions based on their resemblance to reference answers, which might not accurately reflect text quality in scenarios with a vast output space [1]. An emerging alternative involves employing Language Models like GPT-4 for scoring predictions. However, this method is cost-intensive and lacks reproducibility due to unpredictable updates. Considering these evaluation challenges, including generation tasks within this OOD benchmark might not be appropriate.
>
> - For commonsense reasoning, multiple datasets exist from various sources, each demanding distinct knowledge. These differences are substantial enough that knowledge gained from an ID dataset might not be applicable to OOD datasets. For instance, HellaSwag [2] necessitates basic world knowledge and logical reasoning to complete a sentence, while StepGame [3] relies on spatial imagination without requiring world knowledge. The dissimilarity in required abilities suggests that models should acquire knowledge through pre-training rather than fine-tuning on an ID dataset and then transferring it to OOD tasks. This approach aligns more reasonably with the task's nature.
>
> [1] News Summarization and Evaluation in the Era of GPT-3. Goyal et al. 2022.
> [2] HellaSwag: Can a Machine Really Finish Your Sentence? Zellers et al. ACL 2019.
> [3] StepGame: A New Benchmark for Robust Multi-Hop Spatial Reasoning in Texts. Shi et al. AAAI 2022.

---

> > ### Comment · Reviewer_XEXa · 2023-08-29
> >
> > Thanks you for comments.
> >
> > The authors have added additional results and discussion regarding the dataset selection protocol, out-of-distribution (OOD) data definition and the reasons for not choosing NLG tasks. I agree with the authors' explanation of issues like data leakage and novel benchmarks. Although, I find the proposed OOD dataset selection framework novel and interesting, I am not certain if the BOSS benchmark will sustain the test of time, as the datasets are already old and somewhat popular. Due to this, I feel that this benchmark can be used to conduct analysis done in section 4 like analyzing model with different sizes, training steps and in-context learning capabilities, it might not measure the true OOD robustness of large language models due to old or popular datasets.
> >
> > Due to this, I will stick with my initial score of 6.

---

> ### Comment · Area_Chair_yQb7 · 2023-08-29
>
> Dear Reviewer
>
> Kindly review and reply to the feedback provided by the authors.
>
> Regards
> AC

---

### Official Review · Reviewer_5TMS · 2023-07-21
**Review of Revisiting Out-of-distribution Robustness in NLP: Benchmark, Analysis, and LLMs Evaluations**

**Rating:** 6
**Confidence:** 3

**Strengths:**

- The paper provides meaningful evaluation protocol for OOD robustness evaluation that is crucial for training NLP systems.
- Meaningful analysis on three typical learning strategies on OOD robustness.
- Various evaluations on five LLMs that have various sizes and showed experimental results on the number of ID dataset.
- Significant benchmark and evaluation methods on language model's OOD robustness that is helpful for NLP community.
- Showed weaknesses of previous OOD evaluations such as using IMDB and SST.
- Interesting results on different ID-OOD tendency for various types of tasks.

**Additional Feedback:**

- Typos(?): Table 1 IMDB->IMDb for completeness
- Q1: Is SimCSE enough to measure the similarity of the datasets including the text style? (SimCSE seems really good for measuring semantic similarity)
- Q2: What's main reason that the performance of NLI task is significantly dropped when using EDA in Table 5?
- Why "neutural" class is regarded import? (L158)

**Clarity:**

The paper is well written and was not difficult to follow the key concepts.



**Correctness:**

The overall experiments about OOD robustness in the paper look like appropriate and performed correctly.

**Documentation:**

- Enough details to reproduce the results.

**Ethics:**

- No ethical concerns for this paper.

**Limitations:**

- Proposed evaluation protocol BOSS depends on SimCSE when measuring the similarity of the dataset and it maybe not enough. Well, I know that the paper also used the results of OOD performance gap from Table 3 and I just not sure about the ability of SimCSE in measuring domain shift. Are there any references on that?
- The paper only focuses on classification tasks for measuring OOD robustness and do not provide some benchmarks for generation tasks that are also essential and common in NLP tasks.
- To measure the performance among the size of LLMs, it might be used same types of LLM. In other words, it would be better use various sizes of LLaMa. (7B, 13B, 70B)

**Opportunities For Improvement:**

- It would be better to add figure of the entire protocol that is proposed in the paper near Section 2.
- Linear fit may not be the good choice to support the results about relation in Figure 1. (What's the r^2?)

**Relation To Prior Work:**

The paper showed some important limitations of the prior works that focus on OOD robustness evaluation and mentioned them properly. Also, the related work section explained various important prior works on OOD detection with two subsections.

**Summary And Contributions:**

This paper proposes a new evaluation protocol BOSS for out-of-domain (OOD) robustness for various NLP tasks. Also, the paper pointed out the problem in existing evaluation methods for OOD evaluation. And the paper investigates in-domain (ID)-OOD correlation on several categories to probe the causes of these correlations. Finally, the paper validated the training strategies (e.g. full fine-tuning, few-shot fine-tuning) of PLMS on handling OODS (robustness improvement techniques) for various size of LLMs.

---

> ### Author Response · Authors · 2023-08-13
> **Response to Reviewer 5TMS (1/2)**
>
> # Opportunities For Improvement
> 1. It would be better to add figure of the entire protocol that is proposed in the paper near Section 2.
>
> **A**: Thanks for your advice. We will add one figure to explain the protocol more clearly.
>
> 2. Linear fit may not be the good choice to support the results about relation in Figure 1. (What's the r^2?)
>
> **A**: Thanks for the advice. We compute $r^2$ by the following formula:
>
> $ r^2 = \left( \frac{n \sum xy - (\sum x)(\sum y)}{\sqrt{\left[n \sum x^2 - (\sum x)^2\right]\left[n \sum y^2 - (\sum y)^2\right]}} \right)^2 $
>
> For the linear fit presented in Figure 1a, the $r^2$ value is 0.9677. In Figure 1b, the left and right fits have corresponding $r^2$ values of 0.9553 and 0.9396. As for Figure 1c, the $r^2$ values are 0.7690 and 0.8124 for the two fits. These high $r^2$ values signify a strong correlation, thus lending support to our analysis. We will incorporate these values into the revised manuscript.
>
> # Limitations
> 1. I just not sure about the ability of SimCSE in measuring domain shift. Are there any references on that?
>
> **A:** We refer to[1] (Section 3.2) as a representative study measuring text distribution shift via semantic vector distance. While [1] employs CLIP's encoder for multimodal contexts, our NLP-specific context steers us towards SimCSE for semantic representation.
>
> [1] CLIP-ViP: Adapting Pre-trained Image-Text Model to Video-Language Representation Alignment. Xue et al. ICLR 2023.
>
> 2. The paper only focuses on classification tasks for measuring OOD robustness and do not provide some benchmarks for generation tasks that are also essential and common in NLP tasks.
>
> **A:** In our benchmark, we have incorporated Extractive QA as a task not involving classification.
>
> While constructing our benchmark, we did explore the possibility of extending to other tasks like NLG and commonsense reasoning. However, we encountered significant challenges in practical implementation.
>
> -For NLG tasks, the primary concern lies in evaluation. Since different datasets exhibit distinct text styles, an in-domain (ID) model might generate responses with styles diverging from the reference answers when tested on out-of-domain (OOD) datasets. However, current NLG metrics evaluate predictions based on their resemblance to reference answers, which might not accurately reflect text quality in scenarios with a vast output space [2]. An emerging alternative involves employing Language Models like GPT-4 for scoring predictions. However, this method is cost-intensive and lacks reproducibility due to the unpredictable updates. Considering these evaluation challenges, including generation tasks within this OOD benchmark might not be appropriate.
>
> -For commonsense reasoning, multiple datasets exist from various sources, each demanding distinct knowledge. These differences are substantial enough that knowledge gained from an ID dataset might not be applicable to OOD datasets. For instance, HellaSwag [3] necessitates basic world knowledge and logical reasoning to complete a sentence, while StepGame [4] relies on spatial imagination without requiring world knowledge. The dissimilarity in required abilities suggests that models should acquire knowledge through pre-training rather than fine-tuning on an ID dataset and then transferring to OOD tasks. This approach aligns more reasonably with the task's nature.
>
> [2] News Summarization and Evaluation in the Era of GPT-3. Goyal et al. 2022.
> [3] HellaSwag: Can a Machine Really Finish Your Sentence? Zellers et al. ACL 2019.
> [4] StepGame: A New Benchmark for Robust Multi-Hop Spatial Reasoning in Texts. Shi et al. AAAI 2022.

---

> ### Author Response · Authors · 2023-08-13
> **Response to Reviewer 5TMS (2/2)**
>
> # Limitation
> 3. To measure the performance among the size of LLMs, it might be used same types of LLM. In other words, it would be better use various sizes of LLaMa. (7B, 13B, 70B)
>
> **A:** Thanks for your advice. We supplement experiments with LLaMA-13b. But due to computation limits, we are not able to run models that are too large. We use LLaMA-1 since our previous experiments are based on LLaMA-1. The results are as follows:
>
> | Model | Task | SA |  |  |  | TD |  |  |  | NLI |  |  |  |
> |---|---|:---:|:---:|:---:|:---:|:---:|:---:|:---:|:---:|:---:|:---:|:---:|:---:|
> |  | Dataset | AZ | DS | SE | SST | CC | AC | IH | TG | MN | AN | CN | WN |
> | LLaMA-7B | 0-shot | 75.66 | 54.05 | 37.60 | 46.43 | 67.72 | 43.70 | 57.33 | 59.98 | 32.81 | 26.83 | 68.18 | 44.44 |
> |  | ICL | 84.30 | 55.19 | 42.66 | 59.14 | 89.70 | 20.17 | 63.62 | 59.68 | 39.81 | 33.50 | 19.30 | 38.17 |
> |  | ICL* | - | 52.26 | 47.25 | 53.80 | - | - | - | 60.74 | - | 33.47 | - | 37.98 |
> | LLaMA-13B | 0-shot | 81.35 | 56.48 | 42.73 | 59.59 | 89.87 | 19.60 | 62.65 | 58.80 | 32.07 | 24.43 | 47.06 | 38.14 |
> |  | ICL | 82.72 | 54.71 | 40.63 | 57.69 | 83.91 | 38.88 | 66.96 | 67.87 | 38.58 | 34.28 | 20.42 | 38.26 |
> |  | ICL* | - | 46.56 | 38.46 | 53.05 | - | - | - | 68.09 | - | 36.00 | - | 37.31 |
>
> | Model | Task | NER |  |  |  | EQA |  |  |  |
> |---|---|:---:|:---:|:---:|:---:|:---:|:---:|:---:|:---:|
> |  | Dataset | FN | CoNLL | ENER | WNUT | SQuAD | AQA | NQA | SQA |
> | LLaMA-7B | 0-shot | 0.49 | 0.38 | 0.07 | 0.00 | 58.98 | 30.22 | 40.78 | 45.80 |
> |  | ICL | 0.63 | 0.70 | 0.81 | 0.16 | 67.57 | 37.35 | 44.15 | 43.78 |
> |  | ICL* | - | 0.21 | - | 0.00 | - | 37.09 | 48.32 | - |
> | LLaMA-13B | 0-shot | 0.16 | 1.00 | 0.47 | 0.00 | 66.90 | 37.34 | 45.15 | 55.60 |
> |  | ICL | 0.37 | 1.11 | 0.85 | 0.00 | 69.71 | 41.90 | 45.32 | 58.67 |
> |  | ICL* | - | 0.50 | - | 0.00 | - | 40.99 | 49.30 | - |
>
> From the results, we can conclude that simply scaling the model size cannot guarantee the improvement of zero-shot learning and in-context learning.  For example, on NER  and EQA, LLaMA-13B is generally better than LLaMA-7B, while its zero-shot performance on NLI datasets and in-context performance on SA datasets appear to be worse.
>
> # Correctness
> 1. Typos(?): Table 1 IMDB->IMDb for completeness
>
> **A:** Thanks for pointing that out. We would conduct careful proofread in the revision.
>
> 2. Is SimCSE enough to measure the similarity of the datasets including the text style? (SimCSE seems really good for measuring semantic similarity)
>
> **A:** Please kindly refer to the response to Q1. Moreover, apart from employing SimCSE for quantitative similarity assessment, we also undertake qualitative analysis of dataset origins and collection methodologies to ensure clear differentiation.
>
> 3. What's main reason that the performance of NLI task is significantly dropped when using EDA in Table 5?
>
> **A:** We hypothesize that due to the comparatively shorter length of MNLI samples in comparison to Amazon and Civil datasets, NLI models could potentially be more susceptible to perturbations introduced by EDA techniques, like random replacement and random insertion.
>
> 4. Why "neutural" class is regarded import? (L158)
>
> **A:** We give precedence to ternary classification for sentiment analysis due to its greater complexity compared to binary classification. The inclusion of an additional `neutral` class introduces ambiguity into sentiment classification boundaries. Moreover, in practical applications, there are numerous instances involving nonpolar sentiments. Thus, opting for this approach aligns well with real-world complexities.

---

> ### Author Response · Authors · 2023-08-26
> **Hope to receive your reply**
>
> Dear reviewer,
>
> Thanks for your efforts and your insightful suggestions on this paper. We have tried our best to address your concerns and have incorporated your advice to improve the paper. If possible, could you please kindly take a look at our response? Please also let us know if there is any further question about our work, thanks!
>
> Best regards,
>
> Authors

---

> ### Comment · Area_Chair_yQb7 · 2023-08-29
>
> Dear Reviewer
>
> Kindly review and reply to the feedback provided by the authors.
>
> Regards
> AC

---

> > ### Comment · Reviewer_5TMS · 2023-08-31
> >
> > Sorry for the late reply.
> >
> > Thank you for the detailed comments and improvements for the review.
> >
> > I agree with the author's points for the difficulties of evaluation in NLG systems. Also it's really nice to see the results about LLaMa. And thank you all for your comments. However, I am not sure whether [1] is appropriate reference for the reason for using SimCSE. But that doesn't mean that this paper should be rejected, and I think it's a great paper overall.

---

### Author Response · Authors · 2023-08-20
**Summary of the Revision**

Dear Reviewers,

We sincerely thank you for your valuable and constructive comments, and we have updated the paper to incorporate your advice.

To summarize, we have the following updates:
1. We add a figure to demonstrate the dataset selection protocol.
2. We define the "out-of-distribution" considered in this paper in the introduction.
3. We supplement explanations for Takeaway 3 in section 4.2.
4. We clarify some statements that might cause confusion.
5. We add a "Frequently Asked Questions" section in Appendix, explaining why other tasks like NLG and commonsense reasoning are not involved and why we choose SimCSE to measure the distribution shifts.
6. We rearrange the survey table (previous Table 7, current Table 8) to make it clearer.


We are looking forward to hearing from you about any further feedback. If you have any questions or suggestions, we will try our best to address them and improve our paper.

Best,

Authors

---

### Decision · Program_Chairs · 2023-09-22

**Decision:**

Accept (Poster)

**Comment:**

This paper aims to address a very important (and also a very big) problem in NLP. On the one hand, the authors have made significant efforts in dataset analysis, evaluations, and potential solutions. On the other hand, reviewers have raised concerns from many perspectives like the outdated datasets and the potential possibility of data leakage, relatively simple (e.g., sentiment analysis) task, and the limited representations of these few tasks to be claimed as NLP. The use of similarity by a specific model is another valid concern. Overall, there are values in the paper but the authors are encourage to make improvement in terms of paper presentation to avoid the concerns from the readers. It would be considered as a form of over-claim w.r.t. OOD robustness in NLP, if the findings are only supported by the listed datasets and tasks.